# The transcription factor Xrp1 orchestrates both reduced translation and cell competition upon defective ribosome assembly or function

**Marianthi Kiparaki[1,2]\*, Chaitali Khan[1‡†], Virginia Folgado-Marco[1], Jacky Chuen[1], Panagiotis Moulos[2], Nicholas E Baker[1,3,4]\***

[1]Department of Genetics, Albert Einstein College of Medicine, The Bronx, United States; [2]Institute for Fundamental Biomedical Research, Biomedical Sciences Research Center "Alexander Fleming", Vari, Greece; [3]Department of Developmental and Molecular Biology, Albert Einstein College of Medicine, The Bronx, United States; [4]Department of Opthalmology and Visual Sciences, Albert Einstein College of Medicine, The Bronx, United States

**\*For correspondence:** kiparaki@fleming.gr (MK); nicholas.baker@einsteinmed.org (NEB)

**Present address:** [†]Cell and Developmental Biology Center, National Heart Lung and Blood Institute, National Institutes of Health, Bethesda, United States

[‡]This author made a significant contribution

**Competing interest:** The authors declare that no competing interests exist.

**Abstract** Ribosomal Protein (*Rp*) gene haploinsufficiency affects translation rate, can lead to protein aggregation, and causes cell elimination by competition with wild type cells in mosaic tissues. We find that the modest changes in ribosomal subunit levels observed were insufficient for these effects, which all depended on the AT-hook, bZip domain protein Xrp1. Xrp1 reduced global translation through PERK-dependent phosphorylation of eIF2α. eIF2α phosphorylation was itself sufficient to enable cell competition of otherwise wild type cells, but through Xrp1 expression, not as the downstream effector of Xrp1. Unexpectedly, many other defects reducing ribosome biogenesis or function (depletion of TAF1B, eIF2, eIF4G, eIF6, eEF2, eEF1α1, or eIF5A), also increased eIF2α phosphorylation and enabled cell competition. This was also through the Xrp1 expression that was induced in these depletions. In the absence of Xrp1, translation differences between cells were not themselves sufficient to trigger cell competition. Xrp1 is shown here to be a sequence-specific transcription factor that regulates transposable elements as well as single-copy genes. Thus, Xrp1 is the master regulator that triggers multiple consequences of ribosomal stresses and is the key instigator of cell competition.

## Editor's evaluation

This paper will be of broad interest to biologists and has potential clinical relevance. It reveals that defects in ribosome biogenesis or function lead to PERK-phosphorylation of eIF2alpha and render cells vulnerable to cell competition by wild-type neighbors. Cell competition requires induction of the transcription factor Xrp1 in the mutant cell. Thus, the authors demonstrate that Xrp1 is the master regulator of cell competition. A series of compelling experimental manipulations dissecting the epistatic relationship between ribosome defects, Xrp1, eIF2alpha phosphorylation and cell competition support the key claims of the paper.

## Introduction

It would be difficult to exaggerate the importance of ribosomes. Eukaryotic ribosomes comprise 4 rRNAs and 80 proteins combined into Large and Small subunits (LSU and SSU) that, together with

multiple initiation and elongation factors, constitute the translational apparatus for protein synthesis (*Jackson et al., 2010*; *Thomson et al., 2013*). Ribosome biogenesis, and the regulation of translation, are important targets of cellular regulation, and defects affecting ribosomes and translation are implicated in many diseases, from neurodegeneration to cancer (*Aspesi and Ellis, 2019*; *Hetman and Slomnicki, 2019*; *Genuth and Barna, 2018*; *Ingolia et al., 2019*; *Joazeiro, 2019*; *Phillips and Miller, 2020*). Mutations affecting rRNA synthesis, ribosomal protein genes (Rp genes), and some other ribosome biogenesis factors give rise to ribosomopathies, a family of translation-related diseases (*Kampen et al., 2020*). The ribosomopathy Diamond Blackfan Anemia (DBA) most commonly results from heterozygosity for mutations in Rp genes, and is characterized by early onset anemia, cancer predisposition, and sometimes diminished growth and skeletal defects (*Draptchinskaia et al., 1999*; *Choesmel et al., 2007*; *Danilova and Gazda, 2015*; *Da Costa et al., 2018*). Most ribosomal protein genes are also haploinsufficient in *Drosophila*, where their dominant 'Minute' phenotype was named by Bridges and Morgan on account of the small, thin cuticular bristles observed, in addition to developmental delay (*Bridges and Morgan, 1923*; *Lambertsson, 1998*; *Marygold et al., 2007*) .

*Rp* gene loci were recently proposed to be important indicators of aneuploidy (*Ji et al., 2021*). Aneuploid cells can be selectively eliminated from embryonic and developing mammalian tissues, but the mechanisms responsible have been uncertain (*Bolton et al., 2016*; *McCoy, 2017*). In *Drosophila*, cells heterozygous for mutations in *Rp* genes are selectively eliminated from mosaic imaginal discs, where they are replaced by neighboring wild-type cells (*Morata and Ripoll, 1975*; *Simpson, 1979*). This phenomenon, named 'cell competition', represents a process whereby cells that present differences from their neighbors can be eliminated from growing tissues, thought to enable the removal of cells that might be deleterious to the tissue (*Morata and Ripoll, 1975*; *Lawlor et al., 2020*; *Baker, 2020*; *Vishwakarma and Piddini, 2020*; *Marques-Reis and Moreno, 2021*; *Morata, 2021*). Because *Rp* gene dose is likely to be affected whenever one or more chromosomes or substantial chromosome regions are monosomic, cell competition could help eliminate aneuploid cells on the basis of altered *Rp* gene dose (*McNamee and Brodsky, 2009*). This mechanism indeed occurs in *Drosophila* imaginal discs (*Ji et al., 2021*). Such a role of cell competition is potentially significant for tumor surveillance, since tumors almost always consist of aneuploid cells, and for healthy aging, since aneuploid cells accumulate during aging (*Hanahan and Weinberg, 2011*; *López-Otín et al., 2013*). In addition to their mutation in DBA, this provides another reason why it is important to understand the cellular effects of *Rp* mutations, and how they lead to cell competition.

Unsurprisingly, *Rp* mutant heterozygosity generally leads to reduced translation (*Boring et al., 1989*; *Oliver et al., 2004*; *Lee et al., 2018*). It might be expected that a 50% reduction in ribosome subunit biogenesis would be responsible, but remarkably, in *Drosophila* this and many other features of *Rp* haploinsufficiency, including cell competition in the presence of wild-type cells, depend on a bZip, AT-hook putative transcription factor encoded by the *Xrp1* gene (*Lee et al., 2018*). *Xrp1* is responsible for >80% of the transcriptional changes that are seen in *Rp*$^{+/-}$ wing imaginal discs (*Lee et al., 2018*). Thus, reduced translation, which is a feature of Rp haploinsufficiency from yeast to mice and humans, may have a transcriptional basis (*Lee et al., 2018*). Accordingly, we could detect only modest reductions in SSU concentration in heterozygous *RpS3*, *RpS17*, or *RpS18* mutants, although *RpL27A* haploinsufficiency reduced steady state LSU numbers by ~30% (*Lee et al., 2018*). Some of these findings now have support from yeast studies, where deletion of single *Rp* loci present in paralogous pairs (a recent genome duplication has left yeast with many such *Rp* gene pairs) potentially mimics heterozygosity for a single copy gene in diploid organisms. The large majority of translational changes described by ribosome profiling of such yeast pseudo-heterozygotes turned out to reflect changes in mRNA abundance, implicating a predominantly transcriptional response to *Rp* mutations in yeast also (*Cheng et al., 2019*). Mass spectrometry and rRNA measurements of the yeast strains further suggested that ribosome numbers are little affected in most *RpL* gene deletion strains, whereas some *RpS* deletions increase LSU concentrations by up to 1.5 x (*Cheng et al., 2019*). There is also evidence from mice, where it is now suggested that reduced translation in *RpS6*$^{+/-}$ mouse cells depends on the transcription factor p53 (*Tiu et al., 2021*).

These findings raise many mechanistic questions. How does *Rp* haploinsufficiency activate *Xrp1* gene expression?How does this putative transcription factor control overall translation, if not through altered ribosome numbers? Are differences in translation rate between cells the cause of cell competition, or is cell competition due to other consequences of Xrp1 activity?

Alternative views of the *Rp* mutant phenotype have also been presented. Aside from the idea that reduced ribosome levels alter translation directly and are predominantly responsible for human DBA (*Mills and Green, 2017*; *Khajuria et al., 2018*), two recent studies propose that degradation of excess orphan Rp suppresses proteasome and autophagic flux in *Drosophila Rp* mutants, leading to protein aggregation and proteotoxic stress. They propose that proteotoxic stress suppresses translation, and renders *Rp*$^{+/-}$ cells subject to competition with wild-type cells through a further oxidative stress response (*Baumgartner et al., 2021*; *Recasens-Alvarez et al., 2021*). In addition, in concluding that autophagy is protective for *Rp* mutant cells (*Baumgartner et al., 2021*; *Recasens-Alvarez et al., 2021*), these studies contradict previous conclusions that autophagy is only increased in *Rp* mutant cells next to wild-type cells, where it promotes cell death (*Nagata et al., 2019*).

Here, we further investigate the basis of the *Rp* mutant phenotype in *Drosophila*. The results reaffirm the central role of Xrp1 in multiple aspects of the *Rp* mutant phenotype. We confirm the modest effects of *Rp* haploinsufficiency on numbers of mature ribosome subunits, and show directly that ribosome precursors accumulate in *Rp* mutants. We find that translation is reduced in *Rp* mutant cells through eIF2α phosphorylation, but both this and the protein aggregation observed (which appears specific for mutations affecting SSU proteins) require Xrp1 and so are not direct post-transcriptional consequences of ribosome assembly defects, as had been suggested (*Baumgartner et al., 2021*; *Recasens-Alvarez et al., 2021*). We report that interfering with translation, whether through eIF2α phosphorylation or by multiple other routes disrupting ribosome assembly or function, can subject otherwise wild-type cells to competition with normal cells. This is not because translation differences between cells cause cell competition directly, however, but because defects in both ribosome biogenesis and function that affect translation are all found to activate Xrp1, which then mediates the cell competition engendered by these translational stresses. We then show that Xrp1 is a sequence-specific transcription factor that is required for cell competition in response to multiple triggers and is responsible for multiple aspects of the *Rp* mutant phenotype, potentially including transcription of genes that have previously been taken as reporters of oxidative stress. Altogether, these studies clarify discrepancies in previously published work, and refocus attention on transcriptional responses to ribosome and translation defects mediated by Xrp1, with implications for the mechanisms and therapy of multiple ribosomopathies, and for the surveillance of aneuploid cells.

## Results

### Ribosome Levels in *Rp*$^{+/-}$ Cells

Abnormal cellular levels of ribosome subunits has been proposed as the basis for reduced translation in ribosomopathies (*Mills and Green, 2017*). Multiple models of DBA accordingly seek to reduce steady-state Rp concentration to 50% of normal (*Heijnen et al., 2014*; *Khajuria et al., 2018*). By measuring *Drosophila* rRNA levels in northern blots, however, we had previously concluded that whereas cellular levels of ribosome subunits were affected in heterozygotes for an *RpL27A* mutant, multiple *Rp* mutations affecting SSU proteins led only to ~10% reduction in SSU levels that was not statistically significant (*Lee et al., 2018*). A caveat to this conclusion was the use of tubulin mRNA and actin mRNA as loading controls. While mRNA-seq shows that the proportions of actin and tubulin mRNAs are not much affected in *Rp*$^{+/-}$ genotypes (*Kucinski et al., 2017*; *Lee et al., 2018*), it could be that total mRNA amounts are altered by *Rp* mutations, which would affect the conclusions regarding rRNA when mRNA standards are used. In bacteria, it is well-established that ribosomes protect mRNA from turnover, so that reduced ribosome numbers reduce overall mRNA levels as well (*Yarchuk et al., 1992*; *Hui et al., 2014*). The situation in eukaryotic cells may not be the same as in bacteria (*Belasco, 2010*). Still, we decided to measure rRNA levels again using a non-coding RNA as loading control. We chose the 7SL RNA component of Signal Recognition Particle, an abundant non-coding RNA that is expressed in all cells.

Changes in LSU and SSU levels inferred from 5.8 S and 18 S rRNA abundance, normalized to 7SL RNA levels, are shown in *Figure 1*, and a representative northern blot in *Figure 1A*. Similar to what was observed previously, Xrp1 mutations had no effect on apparent LSU or SSU levels in the wild type or in heterozygotes for any of four mutant loci, *RpS18*, *RpS3*, *RpL27A*, and *RpL14*, reaffirming that Xrp1 is unlikely to affect translation rate through an effect on ribosome subunit concentrations (*Figure 1B and C*). Accordingly, *Xrp1*$^{+/+}$ and *Xrp1*$^{+/-}$ data were combined together to compare the

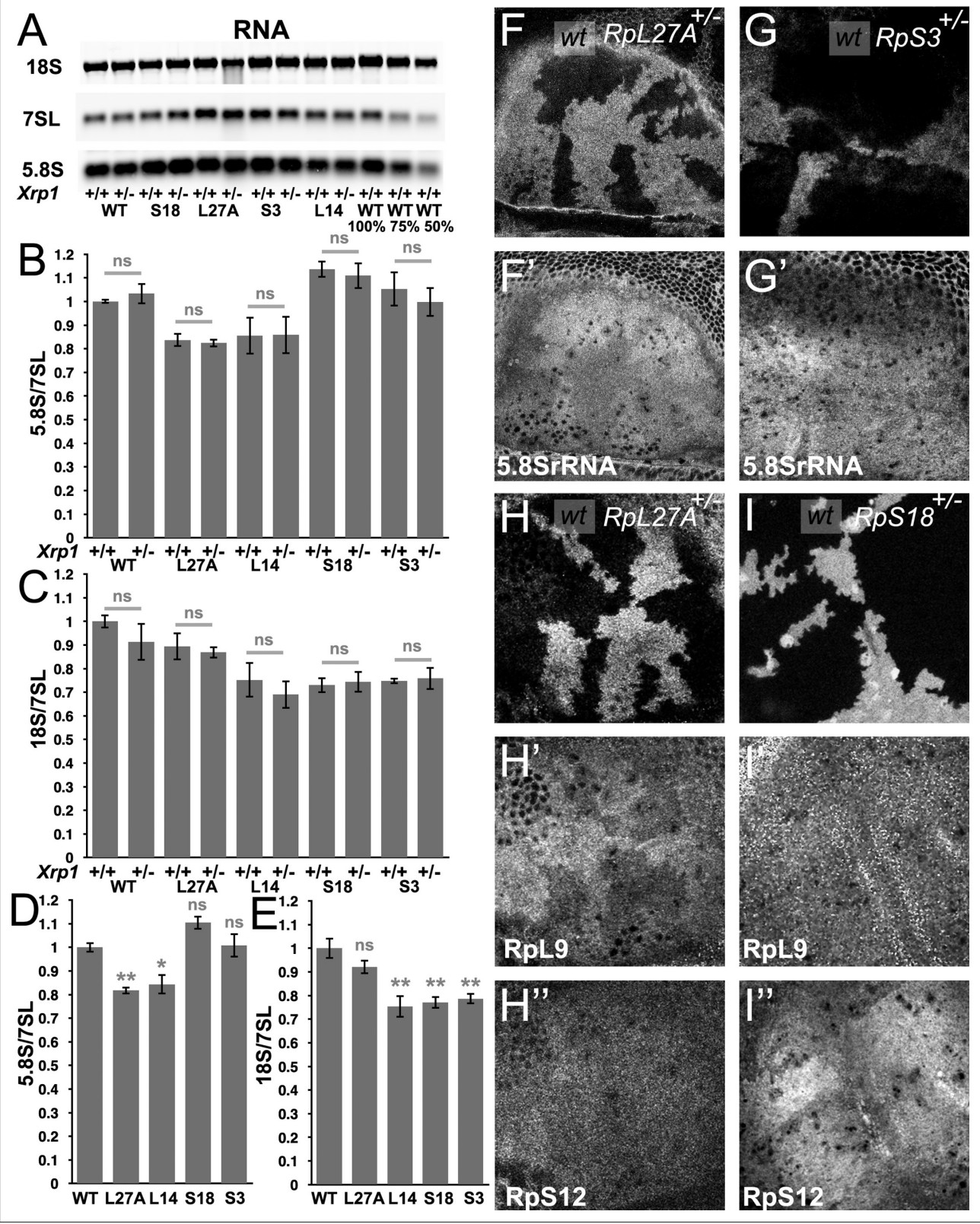

**Figure 1.** Modest changes in ribosomal subunit concentrations in *Rp* mutant wing discs. (**A**) Similar amounts of wing disc RNA from indicated genotypes separated and transferred for northern blotting with, in this case, probes specific for the 18 S rRNA of the ribosomal SSU, the 7SL non-coding RNA for the Signal Recognition Particle, and the 5.8 S rRNA of the ribosomal LSU. Right-most two lanes show serial dilutions of the wild type sample. Panels B-E show signal quantification from multiple such northerns. (**B**) *Xrp1* mutation did not affect LSU concentration in any *Rp* genotype.

*Figure 1 continued on next page*

*Figure 1 continued*

Significance shown only for *Xrp1*[+/+] to *Xrp1*[+/-] between otherwise similar genotypes. Padj values were one in all cases. (**C**) *Xrp1* mutation did not affect SSU concentration in any *Rp* genotype. Significance shown only for *Xrp1*[+/+] to *Xrp1*[+/-] between otherwise similar genotypes. Padj values were one in all cases. (**D**) Two *RpL* mutations reduced LSU concentrations. Significance shown only for comparisons between mutant genotypes and the wild type. Exact Padj values were: 0.00423, 0.0117, 0.0877, 0.858 respectively. (**E**) Two *RpS* mutations, as well as *RpL14*, reduced SSU concentrations. Significance shown only for comparisons between mutant genotypes and the wild type. Exact Padj values were: 0.135, 0.000218, 0.000395, 0.000602 respectively. WT genotype: p{hs:FLP}/w118; p{arm:LacZ} FRT80B/+, Xrp1[+/-] genotype: p{hs:FLP}/w118; FRT82B Xrp1[M2–73]/+, *L27A*[+/-] genotype: p{hs:FLP}/ p{hs:FLP}; *RpL27A*- p{arm:LacZ}FRT40/+; FRT80B/+, *L27A*[+/-]; Xrp1[+/-] genotype: p{hs:FLP}/ p{hs:FLP}; *RpL27A*- p{arm:LacZ}FRT40/+; FRT82B Xrp1[M2–73]/+, *L14*[+/-] genotype: p{hs:FLP}/ p{hs:FLP}; FRT42/+; *RpL14*[1]/+, *L14*[+/-]; Xrp1[+/-] genotype: p{hs:FLP}/ p{hs:FLP}; FRT42/+; *RpL14*[1]/ FRT82 B Xrp1[M2–73]/+, S3[+/-] genotype: p{hs:FLP}/ p{hs:FLP}; FRT82 *RpS3* p{arm:LacZ}/+, S3[+/-]; Xrp1[+/-] genotype: p{hs:FLP}/ p{hs:FLP}; FRT82 *RpS3* p{arm:LacZ}/FRT82B Xrp1[M2–73], *S18*[+/-] genotype: p{hs:FLP}/ p{hs:FLP}; FRT42 *RpS18* p{ubi:GFP} /+; FRT80B/+, *S18*[+/-]; Xrp1[+/-] genotype: p{hs:FLP}/ p{hs:FLP}; FRT42 *RpS18* p{ubi:GFP} /+; FRT82B Xrp1[M2–73]/+ Panels F-I show comparisons between antibody labelings of 5.8 S rRNA, anti-RpL9, or anti-RpS12 between wild type and *Rp*[+/-] cells in mosaic wing imaginal discs. (**F,F'**) *RpL27A* mutation reduced levels of 5.8SrRNA. (**G,G'**) *RpS3* mutation had negligible effect on 5.8 S rRNA levels. (**H,H',H''**) *RpL27A* mutation reduced levels of the LSU component RpL9 but a small effect on the SSU component RpS12. (**I,I',I''**) *RpS18* mutation reduced levels of the SSU component RpS12 but not of the LSU component RpL9. Statistics:One-way Anova with Bonferroni-Holm multiple comparison correction was performed for panels B-E, which were each based on three biological replicates. ns - $p \geq 0.05$.* - $p < 0.05$.** - $p < 0.01$. Genotypes: F, H: p{hs:FLP}/ p{hs:FLP}; RpL27A- p{arm:LacZ} FRT40/FRT40, G: p{hs:FLP}/ p{hs:FLP}; FRT82 *RpS3* p{arm:LacZ} /FRT82B, I: p{hs:FLP}/ p{hs:FLP}; FRT42 *RpS18* p{Ubi:GFP}/FRT42.

The online version of this article includes the following source data and figure supplement(s) for figure 1:

**Source data 1.** Full and unedited blots corresponding to panel A.

**Source data 2.** Northern data underlying panels B-E.

**Figure supplement 1.** Cytoplasmic location of ribosome components.

**Figure supplement 2.** More examples of ribosome levels in *Rp* mutant genotypes.

---

effects of *Rp* mutations. We confirmed that LSU numbers were reduced in the *RpL27A* mutant, and extended this observation to mutation in a second RpL gene, *RpL14* (**Figure 1D**). Unlike our previous study, SSU levels were reduced 20%–30% in *RpS18, RpS3,* and *RpL14* mutants when normalized to the non-coding 7SL RNA, and these reductions were significantly different from the control (**Figure 1E**). By contrast, *RpL27A* did not change SSU numbers (**Figure 1E**).

To confirm these findings using an independent method, we performed tissue staining with a monoclonal antibody, mAbY10B, that recognizes rRNA, and particularly a structure in the 5.8 S rRNA that is part of the LSU (**Lerner et al., 1981**). Consistent with Northern analysis, immunostaining of mosaic wing imaginal discs confirmed lower 5.8 S rRNA levels in *Rp27A*[+/-] cells compared to *Rp27A*[+/+] cells in the same wing discs (**Figure 1F, Figure 1—figure supplement 1A**). By contrast, no reduction in mAbY10B staining was observed in cell mutated for either of two SSU components, *RpS3* or *RpS17*, consistent with the northern blot measurements of 5.8 S rRNA levels (**Figure 1G, Figure 1—figure supplement 1B-D**).

To gain further support for these findings, we compared Rp protein levels by immunostaining mutant and control cells in the same imaginal discs. We used antibodies against RpL10Ab as markers for LSU, and against RpS12 and RACK1 as markers for SSU. *RpL27A* mutant cells contained lower levels of LSU protein, and slightly lower levels of SSU protein (**Figure 1H, Figure 1—figure supplement 2A**). *RpS17*, and *RpS18* mutant cells contained lower levels of the SSU protein, and *RpS18* slightly higher levels of the LSU protein RpL10Ab, even in the Xrp1 mutant background (**Figure 1I, Figure 1—figure supplement 2B-E**). These tissue staining experiments qualitatively support the conclusion that levels of SSU components are generally reduced in *RpS*[+/-] cells, whereas LSU levels were only reduced in *RpL*[+/-] cells (*RpL27A*[+/-]), in comparison to wild type cells within the same preparation, and that these changes are modest and unaffected by Xrp1, even though *Xrp1* mutation restores normal global translation rate (**Lee et al., 2018**).

## Ribosome Precursors Accumulate in *Rp*+/- Cells

An additional, or alternative, potential effect of *Rp* mutations is the accumulation of unused ribosome precursors and assembly intermediates. In yeast, depleting almost any Rp arrests ribosome biogenesis at some stage, reflecting individual requirements for ribosome assembly (**Ferreira-Cerca et al., 2005**; **Ferreira-Cerca et al., 2007**; **Pöll et al., 2009**; **Woolford and Baserga, 2013**; **Henras et al., 2015**). *Rp* haploinsufficiency might delay biogenesis at these same steps, perhaps leading to accumulation

of particular precursor states. To assess ribosome biogenesis in $Rp^{+/-}$ mutants, intermediates were quantified by Northern blotting using probes specific for sequences that are excised from the rRNA as the ribosomes assemble and mature. In *Drosophila*, two parallel pathways A and B excise ITS1, ITS2, and the N-terminal EXT sequences, and process the resulting rRNAs, until the mature 28 S (processed into 28Sa and 28 Sb in *Drosophila*), 18 S and 5.8 S rRNAs are produced by the end of ribosome biogenesis (*Figure 2A*; *Long and Dawid, 1980*). We used specific probes to identify rRNA intermediates on northern blots (*Figure 2A–D*; *Figure 2—figure supplement 1*). As predicted, intermediates accumulated in each of the $Rp^{+/-}$ genotypes (see *Figure 2* legend for details). These findings support the idea that *Rp* gene haploinsufficiency leads to ribosome biogenesis delays, and corresponding accumulation of assembly intermediates. In no case did *Xrp1* mutation eliminate the accumulation of intermediates in *Rp* mutant genotypes (*Figure 2B–D*; *Figure 2—figure supplement 1*). There were some changes noted in the intermediates that accumulated, however. For example, in $RpS17^{+/-}$ and $RpS13^{+/-}$ it seems that more band f accumulates when *Xrp1* is mutated, and less band a (*Figure 2C*).

In mammalian cells with *Rp* haploinsufficiency, unincorporated 5 S RNP, comprising RpL5, RpL11 and the 5 S rRNA, activates the transcription factor and tumor suppressor p53 by inhibiting the p53 ubiquitin ligase MDM2 (*Pelava et al., 2016*). P53 is responsible for at least some consequences of *Rp* haploinsufficiency in mice, perhaps even including the reduction in translation (*Tiu et al., 2021*). P53 is also implicated in cell competition in mammals, although not in *Drosophila*, where Xrp1 may acquire some of its functions (*Kale et al., 2015*; *Baker et al., 2019*). In *Drosophila* it seems that RpS12 is particularly critical for activating Xrp1, through an unknown mechanism (*Kale et al., 2018*; *Lee et al., 2018*; *Boulan et al., 2019*; *Ji et al., 2019*). If a ribosome biogenesis intermediate, which might include RpS12, induced Xrp1 expression, then we predicted that its accumulation and signaling could be prevented by restricting rRNA biogenesis. To test this model, we reduced rRNA synthesis by knockdown of TAF1B, an accessory factor for RNA polymerase I (*Knutson and Hahn, 2011*). We predicted that Xrp1 expression would be reduced when TAF1B was knocked down in an $Rp^{+/-}$ background, and that the knockdown cells would be more competitive than their $Rp^{+/-}$ neighbors. Contrary to these predictions, Xrp1 expression was actually higher in $RpS17^{+/-}$dsRNA$^{TAF1B}$ cells than $RpS17^{+/-}$cells (*Figure 2E*), and $RpS17^{+/-}$ dsRNA$^{TAF1B}$ cells underwent cell death at boundaries with $RpS17^{+/-}$ territories, suggesting they were less competitive, not more so (*Figure 2F*). To understand this result, the effect of TAF1B knockdown in otherwise wild-type cells was examined, and found to resemble that of $RpS17^{+/-}$dsRNA$^{TAF1B}$ cells. That is, dsRNA$^{TAF1B}$ cells strongly activated Xrp1 expression, and underwent apoptosis at interfaces with wild type cells (*Figure 2G, H and J*). This boundary cell death was Xrp1-dependent (*Figure 2I and J*). Thus, far from rRNA being required for Xrp1 expression and cell competition, as expected if an RNP containing RpS12 activates Xrp1, rRNA depletion appeared to have similar effects to Rp depletion.

It has been suggested that Xrp1 might normally be sequestered in nucleoli, only to be released by nucleolar disruption in $Rp^{+/-}$ cells (*Baillon et al., 2018*). We were unable to detect Xrp1 protein sequestered either in nucleoli or elsewhere in wild-type cells, and nucleoli appeared grossly normal in Rp$^{+/-}$ cells by anti-fibrillarin staining, revealing no sign of nucleolar stress (*Figure 2—figure supplement 2A-D*). It is important to compare $Rp^{+/-}$ cells wild-type cells at a level where nuclei are present in both, since in mosaic wing discs $Rp^{+/-}$ nuclei can be displaced basally compared to wild-type cells (eg *Figure 1—figure supplement 1C, D*).

## Reduced protein synthesis is due to PERK-dependent eIF2α phosphorylation in *Rp+/-* cells

*Rp* mutations may lead to surplus unused Rp. In yeast, aggregation of unused Rp rapidly affects specific transcription factors, leading to a transcriptional stress response (*Albert et al., 2019*; *Tye et al., 2019*). To explore how Xrp1 reduces translation, if not through reduced ribosome levels, we investigated the phosphorylation of eIF2α, a key mechanism of global regulation of CAP-dependent translation that responds to proteotoxic stress, among other influences (*Hinnebusch and Lorsch, 2012*). Strikingly, phosphorylation of eIF2α was increased in a cell-autonomous manner in $Rp^{+/-}$ cells compared to $Rp^{+/+}$ cells (RpS3, RpS17, RpS18, and RpL27A were examined) (*Figure 3A and B*; *Figure 3—figure supplement 1A, B*). Control clones lacking Rp mutations did not affect p-eIF2α levels or global translation rate (*Figure 3—figure supplement 1M and N*). Normal p-eIF2α levels were restored in $Rp^{+/-}$ cells, when even one copy of the *Xrp1* gene was mutated, as expected for

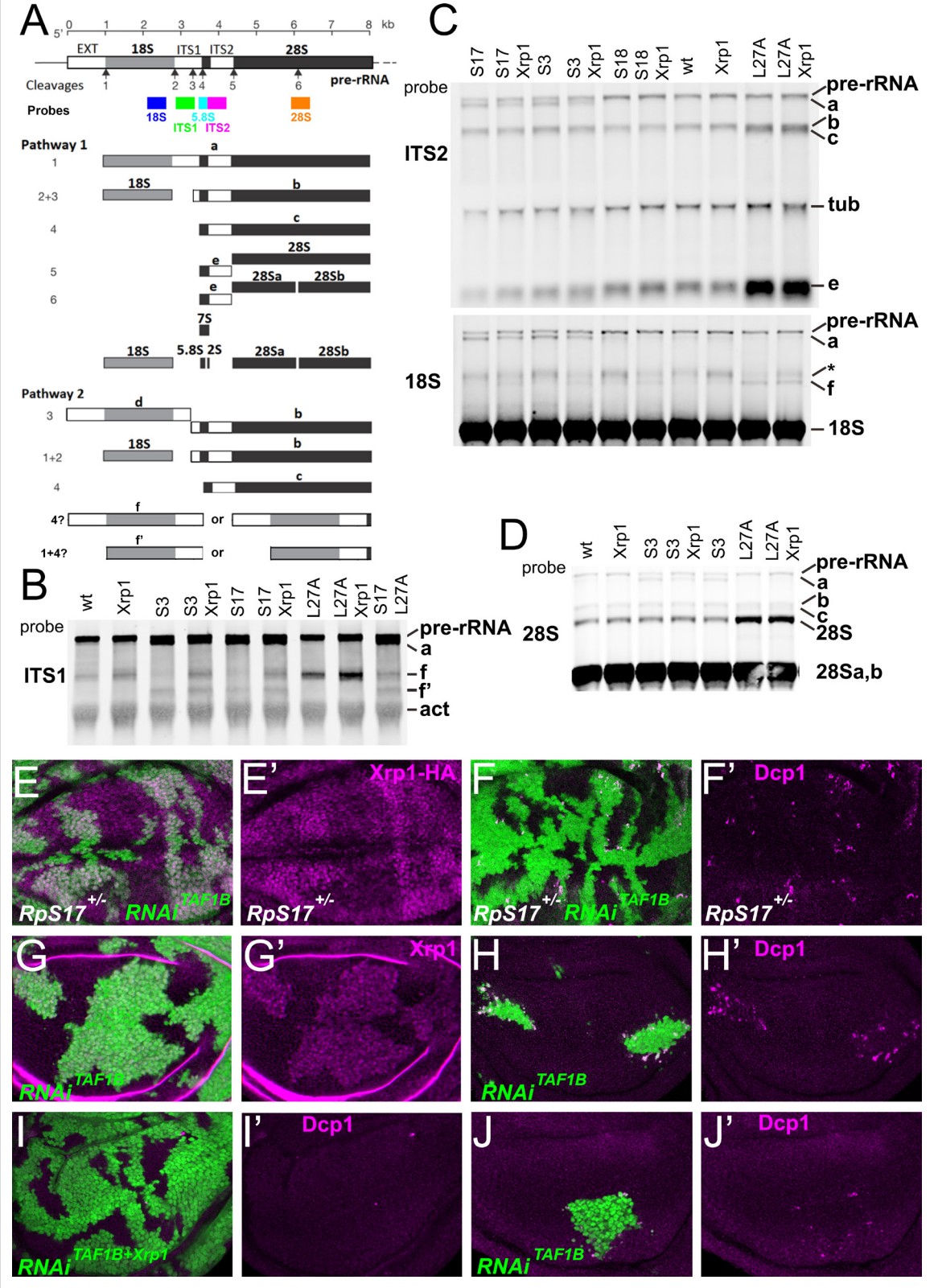

**Figure 2.** Ribosome biogenesis defects and their consequences. (**A**) Two pathways of rRNA processing and the intermediates that result were characterized in *D. melanogaster* embryos by Long and Dawid. Mature 18 S, 5.8 S and 28Sa,b rRNAs are processed from the pre-RNA, along with the removal of two interval sequences ITS1 and ITS2. The cleavages sites were described by Long and Dawid. Colored boxes indicate the probes used in the present study. The 5.8 S probe overlaps with 147 bases at 3'of the ITS1 region, excluding cleavage site 3. Additional intermediates f and **f'** were

*Figure 2 continued on next page*

*Figure 2 continued*

observed in the wing imaginal disc samples. These were recognized by ITS1, 5.8 S (*Figure 2—figure supplement 1*) and 18 S probe and therefore extended beyond the cleavage site 3, although whether beyond site four was uncertain. (**B–D**) Northern blots of total RNA purified from wild-type and $Rp^{+/-}$ wing discs, probed as indicated. (**B**) Reprobed with ITS1 after an initial actin probe. (**C**) Reprobed with ITS2 and then 18 S probes after an initial tubulin probe. Intermediates b, f and the 28 S rRNA (which in *Drosophila* is a precursor to the mature 28Sa and 28 Sb rRNAs) were detected in wild type and $Xrp1^{+/-}$ wing discs, other intermediates only in $Rp^{+/-}$ genotypes. $RpS3^{+/-}$ and $RpS17^{+/-}$ had lower levels of pre-RNA and intermediate (**f**) but accumulate intermediates (**a**) and (**f'**), which might indicate delays in cleavages 2 and 3. $RpS18^{+/-}$ had increased levels of pre-RNA and intermediate (**f**). $RpL27^{+/-}$ accumulated bands (**b**, **c**, **e**, and **f**) and 28 S. The effect on (**f**) suggests crosstalk between RpL27A and SSU processing. (**E–I**) show single confocal planes from mosaic third instar wing imaginal discs. (**E**) TAF1B depletion (green) increased Xrp1-HA levels in $RpS17^{+/-}$ discs (magenta, see also E'). (**F**) TAF1B depletion (green) increased in $RpS17^{+/-}$ discs led to cell death at the boundaries with undepleted cells (active Dcp1 staining in magenta, see also F'). (**G**) TAF1B depletion (green) also increased Xrp1 protein levels in $RpS17^{+/+}$ discs (magenta, see also G'). (**H**) TAF1B depletion (green) led to cell death at the boundaries with undepleted cells (active Dcp1 staining in magenta, see also H'). (**I**) Co-depletion of Xrp1 with TAF1B (green) largely abolished cell death at the clone interfaces (active Dcp1 staining in magenta, see also I'). (**J**) Clones of cells depleted for TAF1B in parallel with panel I, showing reduced clones size and number (green), and competitive cells death at boundaries magenta, see also J'. Additional data related to this Figure is presented in *Figure 2—figure supplement 1*. Genotypes: Northerns: similar to *Figure 1* and additionally: S17$^{+/-}$ genotype: p{hs:FLP}/ p{hs:FLP}; FRT42/+; FRT80 $RpS17$ p{arm:LacZ} /+, S17$^{+/-}$; Xrp1$^{+/-}$ genotype: p{hs:FLP}/ p{hs:FLP}; FRT80 $RpS17$ p{arm:LacZ} /FRT82B $Xrp1^{M2-73}$, S17$^{+/-}$, L27A$^{+/-}$ genotype: p{hs:FLP}/ p{hs:FLP}; RpL27A$^-$ p{arm:LacZ} FRT40/+; FRT80 $RpS17$ p{arm:LacZ} /+, E, F: p{hs:FLP}/+; UAS- RNAi$^{TAF1B}$ /+;$RpS17$, act> CD2> Gal4, UAS-GFP /+ (line: v105873), G, H: p{hs:FLP}/+; UAS- RNAi$^{TAF1B}$ /+;act> CD2> Gal4, UAS- GFP /+ (line: Bl 61957), I: p{hs:FLP}/+; UAS- RNAi$^{TAF1B}$ / UAS-RNAi$^{Xrp1}$;act> CD2> Gal4, UAS- GFP /+ (line: Bl 61957), J: p{hs:FLP}/+; UAS- RNAi$^{TAF1B}$ /TRE-dsRed;act> CD2> Gal4, UAS- GFP /+ (line: Bl 61957) (processed in parallel with 2I).

The online version of this article includes the following source data and figure supplement(s) for figure 2:

**Source data 1.** Full and unedited blots corresponding to panel B.

**Source data 2.** Full and unedited blots corresponding to panel C.

**Source data 3.** Full and unedited blots corresponding to panel D.

**Figure supplement 1.** Additional northern blots detecting rRNA intermediates.

**Figure supplement 1—source data 1.** Full and unedited blots corresponding to *Figure 2—figure supplement 1*.

**Figure supplement 2.** Nucleoli in wild type and *Rp* mutant cells.

---

the Xrp1-dependent process that reduces translation in $Rp^{+/-}$ cells (*Figure 3—figure supplement 1C-E*). To verify that eIF2α regulation by Xrp1 was cell-autonomous, we used clonal knockdown with an *Xrp1* dsRNA previously shown to restore normal growth to $Rp^{+/-}$ cells (*Blanco et al., 2020*). As predicted, *Xrp1* knockdown decreased eIF2α phosphorylation and increased translation rate in a cell-autonomous way (*Figure 3C and D*), as did knocking-down the gene encoding the Xrp1 heterodimer partner, Irbp18 (*Francis et al., 2016*; *Blanco et al., 2020*; *Figure 3—figure supplement 1F, G*).

If eIF2α phosphorylation is how Xrp1 reduces translation in $Rp^{+/-}$ cells, we expected translation to be restored by overexpressed PPP1R15, the *Drosophila* protein homologous to the mammalian p-eIF2α phosphatases, Gadd34 (PPP1R15a) and CReP (PPP1R15b) (*Harding et al., 2009*; *Malzer et al., 2013*). Indeed, PPP1R15 cell-autonomously reduced p-eIF2α levels and cell-autonomously restored overall translation levels in multiple *Rp* genotypes, as measured using the Click reagent o-propargyl puromycin (OPP) (*Figure 3E and F*; *Figure 3—figure supplement 1H, I*). These data indicate that it is eIF2α phosphorylation that suppresses translation in $Rp^{+/-}$ cells.

*Drosophila* contains two potential eIF2α kinases that are thought to respond to particular stresses and not to be activated in unstressed epithelial wing disc cells. When protein kinase R-like endoplasmic reticulum (ER) kinase (PERK), a kinase that phosphorylates eIF2α during the unfolded protein response (*Shi et al., 1998*; *Harding et al., 1999*; *Harding et al., 2000*; *Pakos-Zebrucka et al., 2016*), was depleted using RNAi, p-eIF2α levels were unaffected in wild type wing discs (*Figure 3G*). By contrast, in the $Rp^{+/-}$ genotypes the levels of p-eIF2α were reduced by PERK depletion (*Figure 3H*; *Figure 3—figure supplement 1J, K*). Thus, PERK activity was higher in $Rp^{+/-}$ cells and responsible for eIF2α phosphorylation there. Consistently, PERK knock-down cell-autonomously restored normal translation levels in multiple $Rp^{+/-}$ genotypes (*Figure 3I*; *Figure 3—figure supplement 1L*). Depletion of the other eIF2α kinase known in *Drosophila*, Gcn2, did not decrease p-eIF2α levels in $Rp^{+/-}$ wing disc cells (*Figure 3J*).

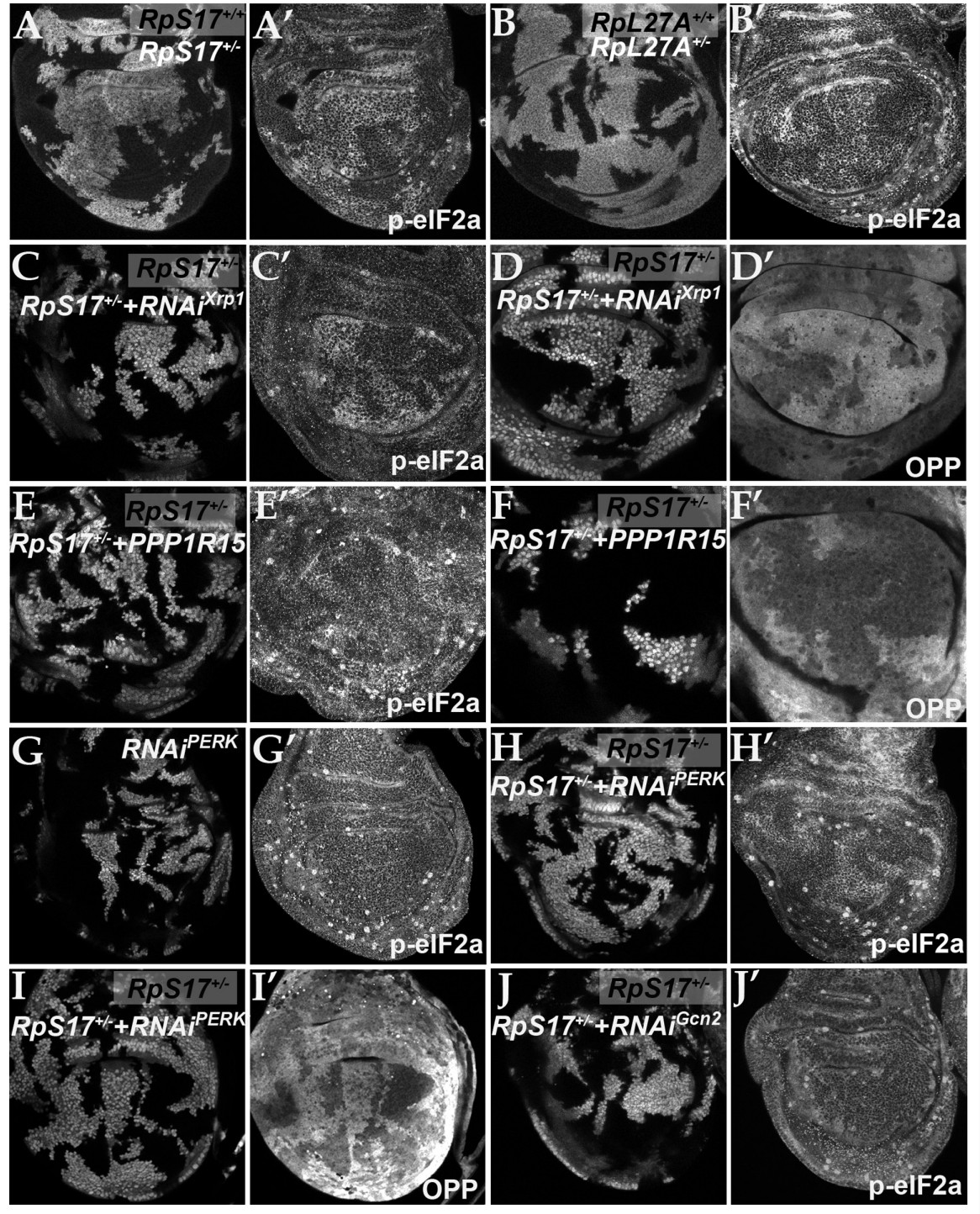

**Figure 3.** eIF2α is phosphorylated in ribosomal protein mutants via Xrp1 and PERK. Panels A-J show single confocal planes from third instar wing imaginal discs. (**A**) Mosaic of *RpS17⁺ᐟ⁻* and *RpS17⁺ᐟ⁺* cells. p-eIF2α levels were increased in *RpS17⁺ᐟ⁻* cells (see A'). (**B**) Mosaic of *RpL27A⁺ᐟ⁻* and *RpL27A⁺ᐟ⁺* cells. p-eIF2α levels were increased in *RpL27A⁺ᐟ⁻* cells (see B'). (**C**) Clones of cells expressing Xrp1-RNAi in a *RpS17⁺ᐟ⁻* wing disc in white p-eIF2α levels were reduced by Xrp1 depletion (see C'). (**D**) Clones of cells expressing Xrp1-RNAi in a *RpS17⁺ᐟ⁻* wing disc in white. Translation rate was restored by Xrp1 depletion (see D'). (**E**) Clones of cells over-expressing PPP1R15 in a *RpS17⁺ᐟ⁻* wing disc in white. p-eIF2α levels were reduced by PPP1R15 over-expression (see E'). (**F**) Clones of cells over-expressing PPP1R15 in a *RpS17⁺ᐟ⁻* wing disc in white. Translation rate was restored by PPP1R15 over-expression (see F'). (**G**) Clones of cells expressing PERK-RNAi in an otherwise wild type wing disc in white. p-eIF2α levels were unaffected (see G'). Note that in this and some other panels mitotic cells are visible near the apical epithelial surface. Mitotic figures, which lack OPP incorporation, are labeled by the anti-p-eIF2α antibody from Thermofisher, but not by the anti-p- eIF2α antibody from Cell Signaling Technologies. (**H**) Clones of cells expressing PERK-RNAi

*Figure 3 continued on next page*

*Figure 3 continued*

in a *RpS17*<sup>+/-</sup> wing disc in whiite. p-eIF2α levels were reduced by PERK knockdown (see H'). (**I**) Clones of cells expressing PERK-RNAi in a *RpS17*<sup>+/-</sup> wing disc in white. Translation rate was restored by PERK knockdown (see I'). (**J**) Clones of cells expressing Gcn2-RNAi in a *RpS17*<sup>+/-</sup> wing disc in white. p-eIF2α levels were not reduced by Gcn2 knockdown (see J'). Further data relevant to this Figure are shown in *Figure 3—figure supplement 1*. Genotypes: A: p{hs:FLP}/+; *RpS17* p{arm:LacZ} FRT80B/FRT80B, B: p{hs:FLP}/ p{hs:FLP}; RpL27A⁻ p{arm:LacZ} FRT40/FRT40, C, D: p{hs:FLP}/+; *RpS17*, act> CD2> Gal4, UAS-GFP /UAS- RNAi<sup>Xrp1</sup>, E,F: p{hs:FLP}/+; UAS-*PPP1R15*/+; *RpS17*, act> CD2> Gal4, UAS-GFP /+, G: p{hs:FLP}/+; UAS- RNAi<sup>PERK</sup> /+;act> CD2> Gal4, UAS-GFP /+, H, I: p{hs:FLP}/+; UAS- RNAiPERK /+; RpS17, act> CD2> Gal4, UAS-GFP /+, J: p{hs:FLP}/+; UAS- RNAi<sup>Gcn2</sup>/+; *RpS17*, act> CD2> Gal4, UAS-GFP /+.

The online version of this article includes the following figure supplement(s) for figure 3:

**Figure supplement 1.** eIF2α phosphorylation in *Rp*<sup>+/-</sup> cell depends on Xrp1 and Irbp18.

## Xrp1 increases protein aggregation and modifies UPR gene expression in *RpS+/-* cells

Recently, protein aggregates have been detected in the cytoplasm of wing disc cells heterozygous for *RpS3*, *RpS23*, and *RpS26* mutants, as foci of ubiquitin or p62 accumulation, reflecting decreased proteasome activity and autophagy (*Baumgartner et al., 2021*; *Recasens-Alvarez et al., 2021*). We confirmed the greater accumulation of aggregates in *RpS3*<sup>+/-</sup>and *RpS18*<sup>+/-</sup>cells compared to wild type cells but did not see this for *RpL27A*<sup>+/-</sup> cells (*Figure 4A–C*). Significantly, another study saw no general increase in autophagy in *RpL14*<sup>+/-</sup> wing discs (*Nagata et al., 2019*). It would be interesting to examine more mutants affecting the LSU, to see whether autophagy is generally unaffected by *RpL* mutations. Importantly, aggregates in *RpS3*<sup>+/-</sup>and *RpS18*<sup>+/-</sup> wing discs were Xrp1-dependent, placing them downstream of Xrp1 activation (*Figure 4D–E*).

PERK is a transmembrane protein with a cytoplasmic kinase domain that is a sensor of unfolded proteins within the ER, not within the cytoplasm or nucleolus ( *Bertolotti et al., 2000*; *Harding et al., 2000*; *Ron and Walter, 2007*; *Walter and Ron, 2011*). Cytoplasmic aggregates can cause unfolded protein accumulation within the ER by competing for proteasomes, however. ER stress also activates Ire-1 and Atf6 in parallel to PERK (*Bertolotti et al., 2000*; *Ron and Walter, 2007*; *Walter and Ron, 2011*; *Hetz, 2012*; *Mitra and Ryoo, 2019*). Xbp1-GFP (*Sone et al., 2013*; *Mitra and Ryoo, 2019*), a reporter for Ire-1 activity, was only inconsistently activated in *Rp*<sup>+/-</sup> wing discs (*Figure 4—figure supplement 1A, C*), in agreement with the absence of any transcriptional signature of Atf6 or Xbp1 activation in *Rp*<sup>+/-</sup> wing disc mRNA-seq data (*Lee et al., 2018*). Crc/Atf4 protein was not upregulated in RpS3<sup>+/-</sup> cells, which would be expected in the classic PERK/ATF4 branch activation of UPR (*Figure 4—figure supplement 1D*). PERK mRNA levels were elevated by 1.4 x in both *RpS3*<sup>+/-</sup> and *RpS17*<sup>+/-</sup> wing discs, however (*Figure 4G*). This increase was statistically very significant, replicated in another group's data, and entirely dependent on *Xrp1* (*Figure 4G*; *Kucinski et al., 2017*; *Lee et al., 2018*). *BiP* and 10 other UPR genes were affected differently. Although none were significantly altered in *RpS17*<sup>+/-</sup> or *RpS3*<sup>+/-</sup> discs, all these genes were affected in *RpS3*<sup>+/-</sup> *Xrp1*<sup>+/-</sup> wing discs, suggesting that Xrp1 prevents their elevation in *RpS17*<sup>+/-</sup> or *RpS3*<sup>+/-</sup> discs (*Figure 4H*). Since these genes help restore ER proteostasis (*Walter and Ron, 2011*), we speculate that Xrp1 might sensitize *Rp*<sup>+/-</sup> cells to PERK activation relative to Atf6 or Xbp1 branches of the UPR (*Lin et al., 2007*), by elevating the expression of PERK while blunting the usual proteostatic response. Testing this notion would require manipulating multiple genes in vivo simultaneously.

## eIF2α phosphorylation is sufficient to induce competitive apoptosis, but through Xrp1

We determined whether manipulating p-eIF2α levels alone was sufficient to cause competition of otherwise wild-type cells. Consistent with this notion, clones of cells depleted for PPP1R15 were rapidly lost from wing imaginal discs and could rarely be recovered (*Figure 5A and B*). Under some conditions (longer heat shock) where clones of cells depleted for PPP1R15 survived temporarily, we verified that p-eIF2α was increased and translation reduced compared to nearby wild-type cells (*Figure 5C and D*; *Figure 5—figure supplement 1A, B*). Such surviving clones were characterized by apoptosis of PPP1R15-depleted cells predominantly at the interface with wild-type cells, a sign of cell competition (*Figure 5E*; *Figure 5—figure supplement 1C*).

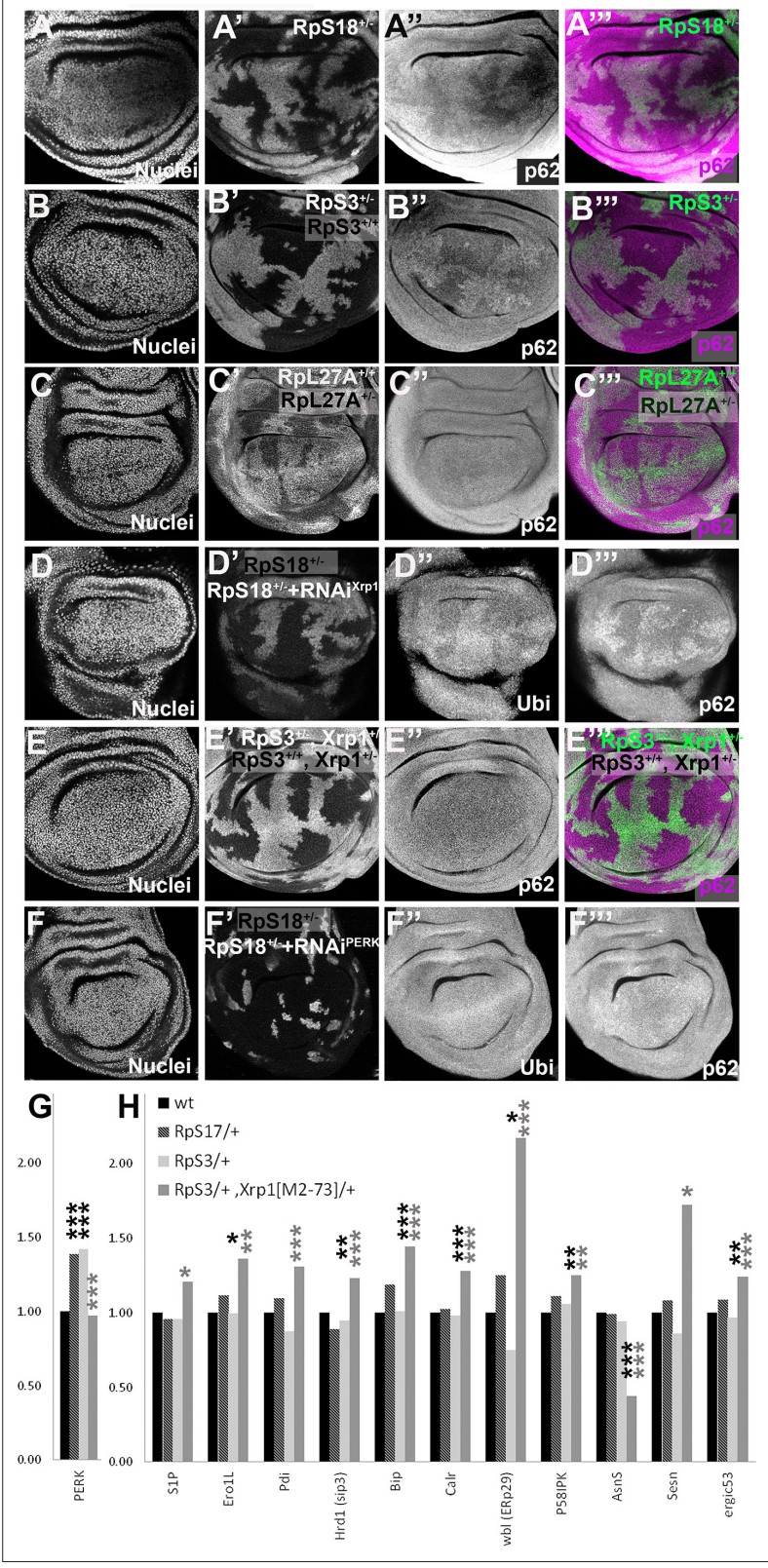

**Figure 4.** Xrp1-dependent aggregates and gene expression changes in *RpS⁺/⁻* cells. Panels A-E show single confocal planes from third instar wing imaginal discs, mosaic for the genotypes indicated. In all cases, the plane passes through the central nuclei-containing disk portion for the genotypes shown. (**A**) p62 was higher in *RpS18⁺/⁻* cells than *RpS18⁺/⁺* cells. (**B**) p62 was higher in *RpS3⁺/⁻* cells than *RpS3⁺/⁻* cells. (**C**) p62 was comparable in *RpL27A⁺/⁻*

*Figure 4 continued on next page*

*Figure 4 continued*

cells and *RpL27A*[+/+] cells. (**D**) Clones of cells expressing Xrp1-RNAi in a *RpS18*[+/-] wing disc in white. Levels of both p62 and ubiquitinylated proteins were reduced by Xrp1 knock-down. (**E**) Mosaic of *RpS3*[+/-] and *RpS3*[+/+] cells in *Xrp1*[+/-] wing disc. No increase in p62 was seen in *RpS3*[+/-] cells (compare panel B). (**F**) Clones of cells expressing PERK-RNAi in a *RpS18*[+/-] wing disc in white. Levels of both p62 and ubiquitinylated proteins remained unaffected by PERK knock-down (**G**). PERK mRNA levels (fold changes in mRNA-seq replicates relative to the wild-type controls according to Deseq2) for the indicated genotypes. PERK mRNA was increased in both *RpS17*[+/-] and *RpS3*[+/-] wing discs but not *RpS3*[+/-], *Xrp1*[M2-73/+] cells. (**H**) mRNA levels for 11 genes participating in the Unfolded Protein Response. All were significantly affected only in the *RpS3*[+/-] *Xrp1* [M2-73/+] genotype. Statistics: Asterisks indicate statistical significance determined by Deseq2 (*: $p_{adj}$ <0.05, **: $p_{adj}$ <0.005, ***: $p_{adj}$ <0.0005) compared to wild type control (black asterisks) or to *RpS3*[+/-] genotype (grey asterisks). Comparisons not indicated were not significant ie $p_{adj}$ ≥0.05 eg *PERK* mRNA in *RpS3*[+/-] *Xrp1* [M2-73/+] compared to wild type control. Further data relevant to this Figure are shown in **Figure 4—figure supplement 1**. Data are based on mRNA-sequencing of 3 biological replicates for each genotype. Genotypes: A: p{hs:FLP}/ p{hs:FLP}; FRT42 *RpS18* p{Ubi:GFP}/FRT42, B: p{hs:FLP}/ p{hs:FLP}; FRT82 *RpS3* p{arm:LacZ} /FRT82B, C: p{hs:FLP}/ p{hs:FLP}; *RpL27A*⁻ p{arm:LacZ} FRT40/FRT40, D: p{hs:FLP}/+; UAS- RNAi[Xrp1]/ GstD lacZ, *RpS18*⁻; act> CD2> Gal4, UAS-GFP /+, E: p{hs:FLP}/ p{hs:FLP}; FRT82 RpS3 p{arm:LacZ} /FRT82B Xrp1[M2-73], F: p{hs:FLP}/+; UAS- RNAi[PERK] / GstD-lacZ, *RpS18*⁻; act> CD2> Gal4, UAS-GFP /+,G-H: wt: w [11-18] /+; FRT82B/+, RpS17/+: w[11-18] /y w p{hs:FLP}; *RpS17* p{ubi:GFP} FRT80B/+, RpS3/+: w[11-18] /y w p{hs:FLP}; FRT82 *RpS3* p{arm:LacZ/+, *RpS3*/+, *Xrp1*[M2-73]/+: w[11-18] /y w p{hs:FLP}; FRT82 *RpS3* p{arm:LacZ}/ FRT82B *Xrp1*[M2-73].

The online version of this article includes the following figure supplement(s) for figure 4:

**Figure supplement 1.** Little UPR detected in *Rp*[+/-] wing discs.

If eIF2α phosphorylation was the downstream effector of Xrp1 that triggers cell competition in *Rp*[+/-] cells then PPP1R15 depletion should eliminate cells independently of Xrp1. Like *Rp*[+/-] cells; however, PPP1R15-depleted cells showed strong upregulation of Xrp1 protein (**Figure 5F and G**; **Figure 5— figure supplement 1D**). When Xrp1 was knocked-down in PPP1R15-depleted cells, competitive cell death was completely blocked, and clone survival improved (Figure H-I; **Figure 5—figure supplement 1E-F**). Even the p-eIF2α levels in the PPP1R15 depleted clones partially depended on Xrp1 (compare **Figure 5C** with **Figure 5J**), and translation rates were similar to wild-type levels in PPP1R15 clones lacking Xrp1 (**Figure 5K**). Interestingly, PPP1R15 knock-down reduced bristle size, another similarity with *Rp* mutants (**Figure 5—figure supplement 2**).

These data raised the possibility of positive feedback between Xrp1 expression and eIF2α phosphorylation. To assess this, we compared Xrp1 expression in *RpS18*[+/-] cells with or without PERK RNAi or PPP1R15 over-expression (**Figure 6A–B**), each of which reduces eIF2α phosphorylation to or below baseline levels (**Figure 3—figure supplement 1H-K**). Because Xrp1 protein levels were unaffected, we concluded that while eIF2α phosphorylation was sufficient to promote Xrp1 expression in otherwise wild-type cells, it was not necessary for the Xrp1 protein expression seen in *RpS18*[+/-] cells (**Figure 6**). This continued Xrp1 expression was functional, because none of Xrp1-dependent JnK activity in *RpS17*[+/-] cells, Xrp1-dependent GstD-LacZ reporter activity in *RpS18*[+/-] cells, or Xrp1-dependent ubiquitin and p62 foci in *RpS18*[+/-] cells were affected by Perk knock-down (**Figure 4**, **Figure 6—figure supplements 1 and 2**). Xrp1 protein levels were reduced by knockdown of its heterodimer partner, Irbp18, in *Rp*[+/-] cells (**Figure 6C**), however. These findings indicate that *Rp*[+/-] cells activate Xrp1 expression independently of eIF2α phosphorylation. Positive feedback between Xrp1 expression and eIF2α phosphorylation might still be important under some circumstances, for example when PPP1R15 is knocked-down, where the effects on global translation and on cell competition depended on Xrp1 (**Figure 5C–E and H–K**).

We also tested whether increased eIF2α phosphorylation was necessary for cell competition (**Figure 5—figure supplement 3**). We used assays where mitotic recombination generates clones of *RpL19*[+/-] cells or clones of *RpL36*[+/-], both subject to competition by surrounding cells (**Figure 5— figure supplement 3A, D**; **Tyler et al., 2007**; **Baillon et al., 2018**). Since PERK was responsible for increasing eIF2α phosphorylation, we expected that if this was required for cell competition, then a *perk* null mutation should rescue the elimination *RpL19*[+/-] or *RpL36*[+/-] clones. Since no *RpL19*[+/-] *perk*[-/-] or *RpL36*[+/-] *Perk*[-/-] clones were recovered (**Figure 5—figure supplement 3B, E**), although *RpL36*[+/+] *perk*[-/-]clones survived normally (**Figure 5—figure supplement 3C, F**), we concluded that PERK-dependent eIF2α phosphorylation was not required for cell competition.

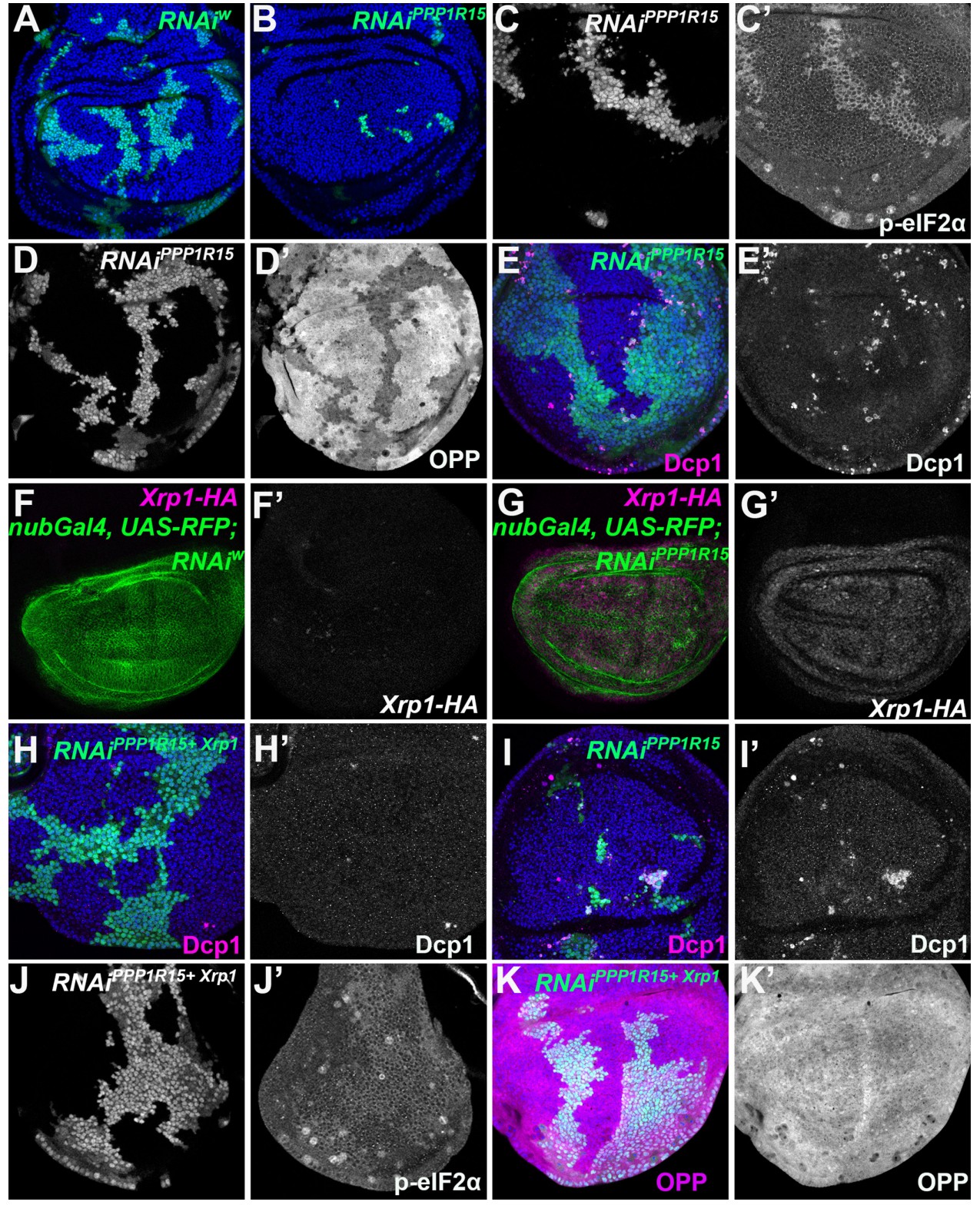

**Figure 5.** eIF2α phosphorylation can induce Xrp1 expression and cell competition. All panels show single confocal planes from third instar wing imaginal discs, mosaic for the genotypes indicated. All the sections pass through the central region of the disc proper containing nuclei in all genotypes, as indicated by the DNA stain in blue in some panels. (**A**) Clones expressing *white* RNAi (green). Clones induced by 7 min heat shock. (**B**) Clones expressing *PPP1R15* RNAi (green)were fewer and smaller than the control (compare panel A). Clones induced by 7 min heat shock. (**C**) Clones expressing

*Figure 5 continued on next page*

*Figure 5 continued*

*PPP1R15* RNAi (white) contain phosphorylated eIF2α (see C'). (**D**) Clones induced by 25 ± 5 min heat shock, which results in larger clone areas (white). Labelled clones expressing *PPP1R15* RNAi reduced translation rate (see D'). (**E**) Labelled clones expressing *PPP1R15* RNAi (green) underwent competitive apoptosis at interfaces with wild type cells (activated caspase Dcp1 in magenta; see also E'). (**F**) *Nub-Gal4* drives gene expression in the wing pouch, shown in green for RFP, with little expression of Xrp1-HA (magenta; see also F'). (**G**) *PPP1R15* RNAi induces Xrp1-HA expression in the wing pouch (magenta; see also G'). (**H**) Clones co-expressing *PPP1R15* RNAi and *Xrp1* RNAi (green) lacked competitive apoptosis (activated caspase Dcp1 in magenta; see also H'). (**I**) Clones expressing *PPP1R15* RNAi (green). Experiment performed in parallel to panel H. Note competitive apoptosis at interfaces with wild type cells (activated caspase Dcp1 in magenta; see also I'), and smaller clone size. Cell death at the basal surface of the same disc shown in ***Figure 5—figure supplement 1F***. (**J**) Clones co-expressing *PPP1R15* RNAi and *Xrp1* RNAi (white) showed less eIF2α phosphorylation than for *PPP1R15* RNAi alone (compare panel C). Sample prepared in parallel to panel C (in the same tube from fixation to staining). (**K**) *Xrp1* knockdown restored normal translation rate to cell clones expressing *PPP1R15* RNAi (green; see also K'). Sample prepared in parallel to panel D (in the same tube from fixation to staining). Additional data relevant to this Figure is shown in ***Figure 5—figure supplement 1***. Genotypes: A: {hs:FLP}/+; act> CD2> Gal4, UAS-GFP / UAS – RNAi^w, B: {hs:FLP}/+; act> CD2> Gal4, UAS-GFP / UAS – RNAi^PPP1R15 (line: BL 33011) (samples were processed on the same day, not on the same tube), C: {hs:FLP}/+; UAS – RNAi^PPP1R15 /TRE-dsRed; act> CD2> Gal4, UAS-GFP /+(line: v107545) (processed in parallel with 5 J), D: {hs:FLP}/+; act> CD2> Gal4, UAS-GFP / UAS – RNAi^PPP1R15 (line: BL 33011), E: {hs:FLP}/+; UAS – RNAi^PPP1R15 /+; act> CD2> Gal4, UAS-GFP /+ (line: v107545),F: nubGal4, UAS-RFP/+; Xrp1-HA/RNAi^w, G: nubGal4, UAS-RFP/ UAS – RNAi^PPP1R15; Xrp1-HA/+ (line: v107545), H, J, K: {hs:FLP}/+; UAS – RNAi^PPP1R15 / UAS-RNAi^Xrp1; act> CD2> Gal4, UAS-GFP /+ (line: v107545) (5 H processed in parallel with 5I. Also, 5 K processed in parallel with ***Figure 5—figure supplement 1B***) (line RNAi^PPP1R15: v107545 and line RNAi^Xrp1: v107860), I: {hs:FLP}/+; UAS – RNAi^PPP1R15 /TRE-dsRed; act> CD2> Gal4, UAS-GFP /+ (line RNAi^PPP1R15: v107545).

The online version of this article includes the following figure supplement(s) for figure 5:

**Figure supplement 1.** eIF2α phosphorylation induces Xrp1 expression and cell competition.

**Figure supplement 2.** eIF2α phosphorylation reduces bristle size.

**Figure supplement 3.** eIF2α phosphorylation is not required for cell competition.

These data show that eIF2α phosphorylation was sufficient to reduce cell competitiveness in otherwise wild type cells, but only in the presence of Xrp1. It was the mechanism whereby Xrp1 reduced global translation rate in $Rp^{+/-}$ mutant cells, but apparently not the downstream effector of Xrp1 for cell competition.

## Interrupting the translation cycle activates Xrp1-dependent cell competition, independently of diminished translation

Phosphorylation of eIF2α inhibits CAP-dependent initiation. To explore further whether reduced translation was sufficient to cause cell competition, we also reduced translation by clonal depletion of translation factors acting at a variety of steps in the translation cycle, not only at initiation but also the 40 S-60S subunit joining and elongation steps (***Jackson et al., 2010***). Specifically, we depleted eIF4G, eIF5A, eIF6, eEF1α1, and eEF2, none of which is encoded by a haploinsufficient gene (***Marygold et al., 2007***). eIF4G is part of the eIF4 complex which binds the mRNA 5'cap and recruits SSU to enable translation initiation (***Jackson et al., 2010***). It is now accepted that eIF5A functions in translation elongation and termination (***Saini et al., 2009***; ***Schuller et al., 2017***). eEF1α1 delivers aminoacyl-tRNAs to the ribosome and eEF2 also promotes ribosome translocation (***Dever and Green, 2012***). eIF6 has a role during LSU biogenesis and also in translation initiation (***Brina et al., 2015***).

All these depletions exhibited severe reduction in translation rate in the third instar larvae, as did TAF1B depletion (***Figure 7A, E, I and M***; ***Figure 7—figure supplement 1A, E***; the fact that clones of cells expressing these dsRNAs could be recovered with such low translation suggests that translation factor depletion probably exacerbates over time, initially being insufficient to prevent translation and growth, but eventually becoming severe). Importantly, all these translation factor depletions resulted in more dramatic induction of apoptosis in depleted cells that were close to wild-type cells than within the clones, suggesting that differences in translation rate might be sufficient to initiate cell competition (***Figure 7B, F and J***; ***Figure 7—figure supplement 1B, F***; ***Figure 7—figure supplement 2***). Interestingly, in all these cases translation increased in wild-type cells near to the affected clones, something that was rare adjacent to $Rp^{+/-}$ cells and not seen adjacent to cells depleted for PPP1R15, although it was observed near to TAF1B depleted cells (***Figure 7A, E, I and M***; ***Figure 7—figure supplement 1A, E***). Phosphorylated RpS6 accumulated in wild-type cells adjacent to TAF1B depleted cells, suggesting that a non-autonomous activation of Tor accounts for the increased translation in cells nearby those with translation deficits (***Figure 7N***; ***Laplante and Sabatini, 2012***; ***Romero-Pozuelo et al., 2017***).

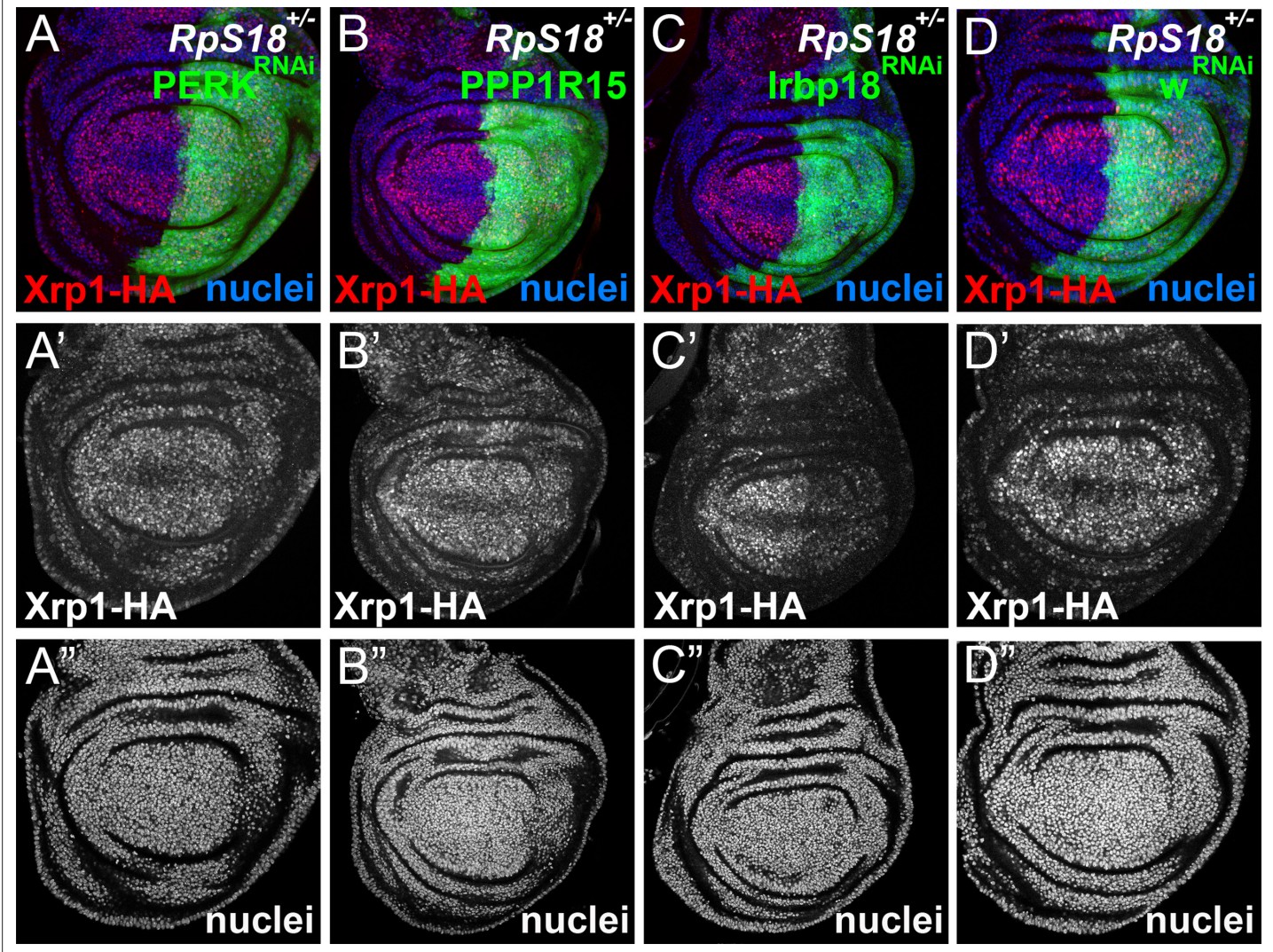

**Figure 6.** eIF2α phosphorylation is dispensable for Xrp1 expression in Minutes. All panels show single confocal planes from *RpS18⁺ᐟ⁻* third instar wing imaginal discs, co-expressing GFP and the indicated constructs in the posterior compartments. All the sections pass through the central region of the disc proper containing nuclei, as indicated by the DNA stain in blue. (**A**) Perk knock-down had no effect on Xrp1 expression in *RpS18⁺ᐟ⁻*. (**B**) PPP1R15 over-expression had no effect on Xrp1 expression in *RpS18⁺ᐟ⁻*. (**C**) Irbp18 knock-down strongly reduced Xrp1 expression in *RpS18⁺ᐟ⁻*. (**D**) Knock-down for the *w* gene had no effect on Xrp1 expression in *RpS18⁺ᐟ⁻*. Genotypes: A: *RpS18⁻*, en-GAL4, UAS-GFP / UAS- RNAi^PERK^; Xrp1-HA /+, B: *RpS18⁻*,en-GAL4, UAS-GFP / UAS-PPP1R15; Xrp1-HA /+, C: *RpS18⁻*,en-GAL4, UAS-GFP /+; Xrp1-HA /UAS- RNAi^Irbp18^, D:*RpS18⁻*,en-GAL4, UAS-GFP /+; Xrp1-HA / UAS- RNAi^w^.

The online version of this article includes the following figure supplement(s) for figure 6:

**Figure supplement 1.** PERK is dispensable for Xrp1-dependent JnK activity in Minute cells.

**Figure supplement 2.** PERK is dispensable for Xrp1-dependent GstD1 activity in Minute cells.

To confirm that translation factor depletion affected translation directly, and downstream of Xrp1 and PERK, Xrp1 expression and eIF2α phosphorylation were examined. Unexpectedly, depletion for translation factors was associated with both cell-autonomous induction of Xrp1 expression and eIF2α phosphorylation (*Figure 7C, D, G, H, K and L*; *Figure 7—figure supplement 1C*,D,G,H; *Figure 7—figure supplement 3*). The levels were at least comparable to those of TAF1B-depleted cells (*Figure 7—figure supplement 1I, J*). When Xrp1 was knocked-down, PPP1R15 overexpressed, or PERK depleted simultaneously with translation factor depletion, the translation factor depletions behaved similarly to one another, and also similarly to TAF1B knock-down. PPP1R15 overexpression reduces eIF2α phosphorylation even in wild type cells, without increasing their global translation

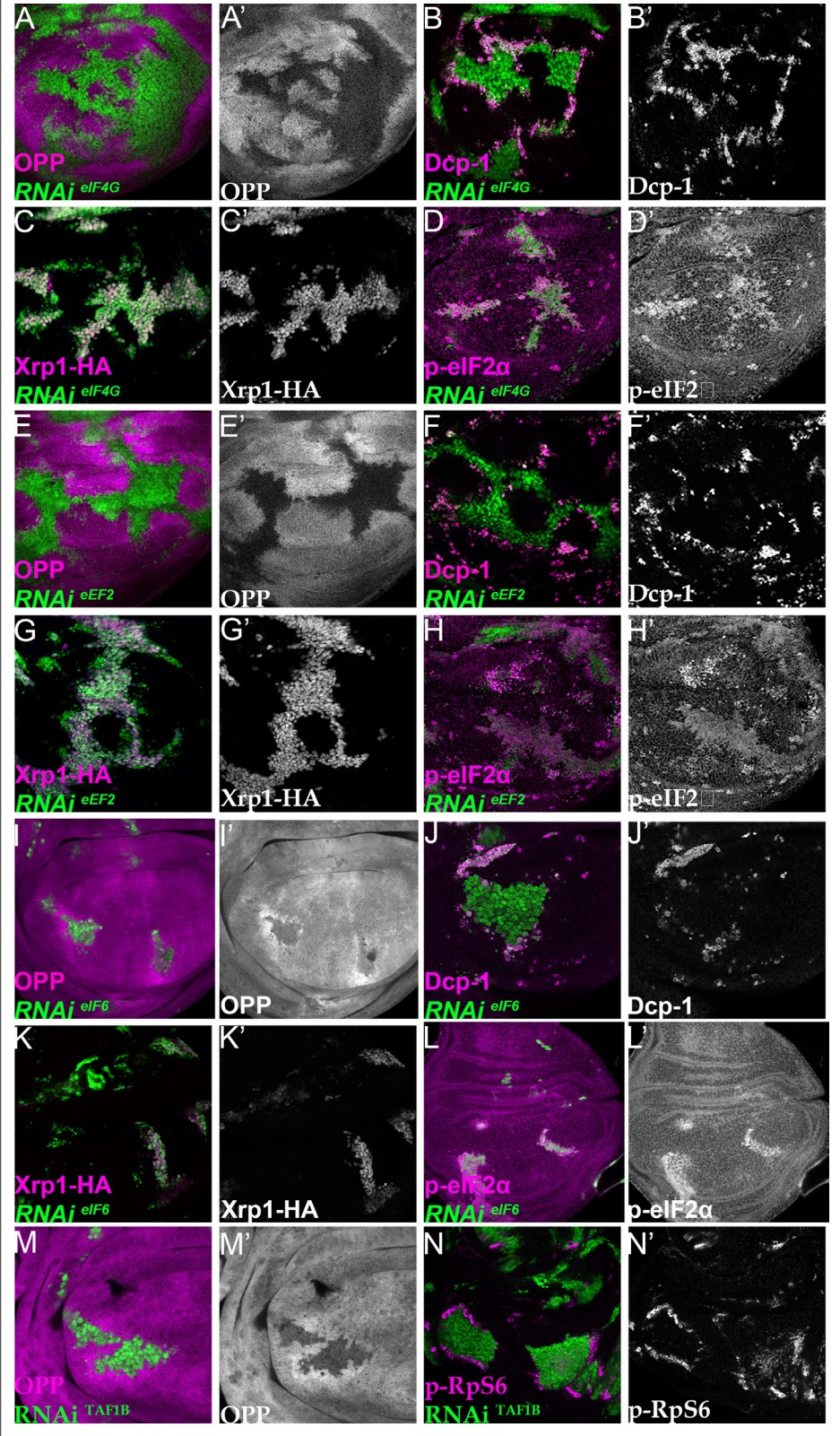

**Figure 7.** Depletion of translation factors induces Xrp1 expression, eIF2α phosphorylation, reduced translation, and cell competition. Clones of cells depleted for translation factors are labelled in green. In each case, translation factor depletion reduced translation rate, resulted in competitive cell death at interfaces with wild type cells, induced Xrp1-HA expression, and led to eIF2α phosphorylation. Translation rate, dying cells (activated caspase

*Figure 7 continued on next page*

*Figure 7 continued*

Dcp1), Xrp1-HA and p-eIF2α are indicated in magenta and in separate channels as labelled. To clarify cell-autonomy, cell death is also shown in higher magnification in *Figure 7—figure supplement 2*. (**A–D**) Clones expressing RNAi for eIF4G. (**E–H**) Clones expressing RNAi for eEF2. (**I–L**) Clones expressing RNAi for eIF6. In all cases (panels A,E,I), wild-type cells near to cells depleted for translation factors show higher translation rate than other wild type cells. (**M**) Clones of cells depleted for TAF1B (green) also showed a cell-autonomous reduction in translation rate and non-autonomous increase in nearby wild-type cells (translation rate in magenta, see also M'). (**N**) Clones of cells depleted for TAF1B (green) showed a non-autonomous increase in RpS6 phosphorylation in nearby cells (magenta, see also N'). Additional data relevant to this Figure is shown in *Figure 7—figure supplement 1* and *Figure 7—figure supplement 2*. Genotypes: A, B, D: {hs:FLP}/+; UAS – RNAi^eIF4G /+; act> CD2> Gal4, UAS-GFP /+ (line: v17002), C:{hs:FLP}/+; UAS – RNAi^eIF4G /+; act> CD2> Gal4, UAS-GFP /Xrp1-HA (line: v17002), E, F, H: {hs:FLP}/+; UAS – RNAi^eEF2 /+; act> CD2> Gal4, UAS-GFP /+ (line: v107268), G: {hs:FLP}/+; UAS – RNAi^eEF2 /+; act> CD2> Gal4, UAS-GFP / Xrp1-HA (line: v107268), I, J, L:{hs:FLP}/+; UAS – RNAi ^eIF6 /+; act> CD2> Gal4, UAS-GFP / + (line: v108094), K: {hs:FLP}/+; UAS – RNAi^eIF6 /+; act> CD2> Gal4, UAS-GFP / Xrp1-HA(line: v108094), M, N: p{hs:FLP}/+; UAS-RNAi^TAF1B/+;act> CD2> Gal4, UAS- GFP /+ (line: Bl 61957).

The online version of this article includes the following figure supplement(s) for figure 7:

**Figure supplement 1.** Xrp1 expression, eIF2α phosphorylation, reduced translation, and cell competition after depletion of additional translation factors.

**Figure supplement 2.** Cell-autonomy of cell death after translation factor depletion.

**Figure supplement 3.** Translation factor knock-down induces Xrp1 expression.

---

rate or affecting survival (*Figure 8—figure supplement 1A-F*). In translation-factor-depleted cells, PPP1R15 overexpression also reduced eIF2α phosphorylation to or even below control levels (*Figure 8A, D and G*, *Figure 8—figure supplement 1G, J and M*, *Figure 8—figure supplement 4*), but this did not restore normal translation rates (*Figure 8B, E and H*, *Figure 8—figure supplement 1H, K and N*). There was no rescue of competitive cell death (*Figure 8C, F and I*; *Figure 8—figure supplement 1I, L and O*) or Xrp1 expression (*Figure 8J–L*; *Figure 8—figure supplement 1P, Q and R*). PERK knock-down similarly did not affect Xrp1 expression or rescue competitive cell death in translation-factor knock-downs or TAF1B knock-down (*Figure 8—figure supplement 2*). Knockdown of Xrp1 reduced levels of eIF2α phosphorylation in some cases (*Figure 8M, P and S Figure 8—figure supplement 3J*), although for eIF5A and eEF1α1 the reduction was only partial so that both the eIF5A Xrp1 depleted and eEF1α1 Xrp1-depleted cells retained more eIF2α phosphorylation than wild-type cells (*Figure 8—figure supplement 3D, G*). For all the translation factors, however, Xrp1 depletion eliminated or strongly reduced cell death at the competing cell boundaries, irrespective of whether eIF2α phosphorylation remained (*Figure 8O, R and U*; *Figure 8—figure supplement 3F, I*). We also found that overall translation rate, as estimated by OPP incorporation, was only partially restored by simultaneous Xrp1 depletion from most translation factor knock-down cells, and remained lower than wild type cells (*Figure 8N and Q*; *Figure 8—figure supplement 3H*). Remarkably, simultaneous knock-down of Xrp1 along with eIF6 resulted in translation rates similar to or higher than in wild type cells (*Figure 8T*). We have also seen this with eEF1α Xrp1 double knockdown (*Figure 8—figure supplement 3E*), but interpretation is difficult because some clones depleted only for eEF1α also had higher OPP labeling. Reduced translation upon TAF1B knock-down was also Xrp1-dependent (*Figure 8—figure supplement 3K, L*), although Xrp1 depletion had no effect on eIF2α phosphorylation, global translation, or cell survival of otherwise wild-type cells (*Figure 8—figure supplement 3A-C*).

These results unexpectedly show that translation factor or polI depletion triggers similar effects to depletion of ribosome components in *Rp* mutants, in which Xrp1 expression leads to eIF2α phosphorylation and to cell competition. The results separate eIF2α phosphorylation from cell competition, however, because Xrp1-dependent competitive cell death continued even when eIF2α phosphorylation levels was restored to normal by PPP1R15 overexpression, and because remaining eIF2α phosphorylation in eIF5A Xrp1-depleted and eEF1α1 Xrp1-depleted cells did not lead to cell competition. The results also separate cell competition from differences in translation levels, because no competitive cell death was observed in eIF4G Xrp1-depleted, eIF5A Xrp1-depleted, and eEF2 Xrp1-depleted cells, even though their translation was lower than the nearby wild type cells. Indeed, depletion for eIF6 or TAF1B induced Xrp1 and cell competition, even though without Xrp1 these cells seemed to

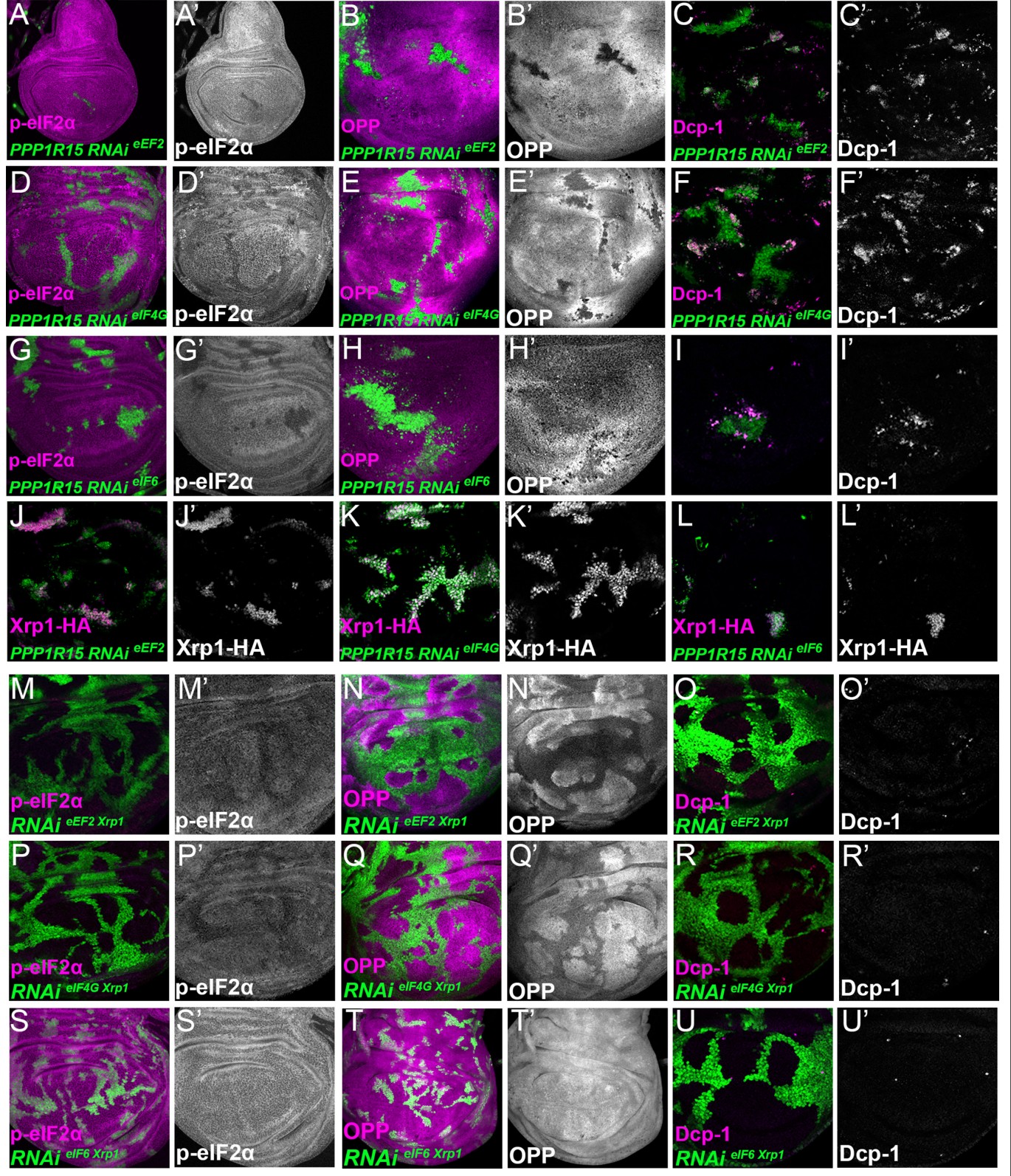

**Figure 8.** Interrupting the translation cycle activates Xrp1-dependent cell competition, independently of diminished translation. Single confocal planes from third instar wing imaginal discs. p-eIF2α levels, translation rate (ortho-propargyl puromycin), dying cells (activated caspase Dcp1) and Xrp1-HA are indicated in magenta and in separate channels as labelled. (**A–L**) Clones of cells depleted for translation factors which also overexpress PPP1R15 are shown in green. In each case, PPP1R15 overexpression was sufficient to reduce eIF2α phosphorylation to near control levels (or even lower), but it did

*Figure 8 continued on next page*

*Figure 8 continued*

not restore normal translation rates, did not affect Xrp1-HA levels and did not reduce competitive cell death. (**A–C**) Clones co-expressing PPP1R15 and RNAi for eEF2. (**D–F**) Clones co-expressing PPP1R15 and RNAi eIF4G. (**G-I**) Clones co-expressing PPP1R15 and RNAi for eIF6. (**J–K**) Xrp1-HA expression (magenta) in clones co-expressing PPP1R15 and RNAi for eEF2 (**J**), eIF4G (**K**), or eIF6 (**L**). (**M–U**) Clones of cells depleted for translation factors which also express Xrp1-RNAi are shown in green. (**M–O**) Clones depleted for Xrp1 as well as eEF2 expressed phospho-eIF2α at near to control levels, only partially restored overall translation rate, but lacked competitive cell death. (**P–R**) Clones depleted for Xrp1 as well as eIF4G expressed phospho-eIF2α at near to control levels, only partially restored overall translation rate, but lacked competitive cell death. (**S–U**) Clones depleted for Xrp1 as well as eIF6 expressed phospho-eIF2α at near to control levels, restored overall translation rate and lacked competitive cell death. Genotypes: A-C: {hs:FLP}/+; UAS – RNAi$^{eEF2}$/ UAS-PPP1R15; act> CD2> Gal4, UAS-GFP / +, D-F: {hs:FLP}/+; UAS – RNAi$^{eIF4G}$ /UAS-PPP1R15; act> CD2> Gal4, UAS-GFP / +, G-I: {hs:FLP}/+; UAS – RNAi$^{eIF6}$/ UAS-PPP1R15; act> CD2> Gal4, UAS-GFP / +, J: {hs:FLP}/+; UAS – RNAi$^{eEF2}$/ UAS-PPP1R15; act> CD2> Gal4, UAS-GFP / Xrp1-HA, K: {hs:FLP}/+; UAS – RNAi$^{eIF4G}$ /UAS-PPP1R15; act> CD2> Gal4, UAS-GFP / Xrp1-HA, L: {hs:FLP}/+; UAS – RNAi$^{eIF6}$/ UAS-PPP1R15; act> CD2> Gal4, UAS-GFP / Xrp1-HA, M-O: {hs:FLP}/+; UAS – RNAi$^{eEF2}$/ UAS- RNAi$^{Xrp1}$; act> CD2> Gal4, UAS-GFP /+, P-R: {hs:FLP}/+; UAS – RNAi$^{eIF4G}$ /UAS- RNAi$^{Xrp1}$; act> CD2> Gal4, UAS-GFP / +, S-U: {hs:FLP}/+; UAS – RNAi$^{eIF6}$/ UAS- RNAi$^{Xrp1}$; act> CD2> Gal4, UAS-GFP / +.

The online version of this article includes the following figure supplement(s) for figure 8:

**Figure supplement 1.** Translation and transcription factor knock down affect global translation, cell competition, and Xrp1 expression, independently of PPP1R15 over-expression.

**Figure supplement 2.** Translation and transcription factor knock down affect Xrp1 expression, and cell survival, independently of PERK knock-down.

**Figure supplement 3.** Role of Xrp1 in eIF2α phosphorylation and global translation after knockdown of translation or transcription factors.

**Figure supplement 4.** Quantification of p-eIF2α levels.

translate at similar or higher rates to their neighbors. These results focus attention on Xrp1 as the key effector of cell competition, irrespective of eIF2α phosphorylation and overall translation rate.

These results also raise the question of whether *Rp* haploinsufficiency, rRNA depletion, eIF2α phosphorylation, and translation factor depletion all activate Xrp1 through a common pathway. In *Rp*$^{+/-}$ genotypes, Xrp1 expression depends on a specific ribosomal protein, RpS12, and is almost

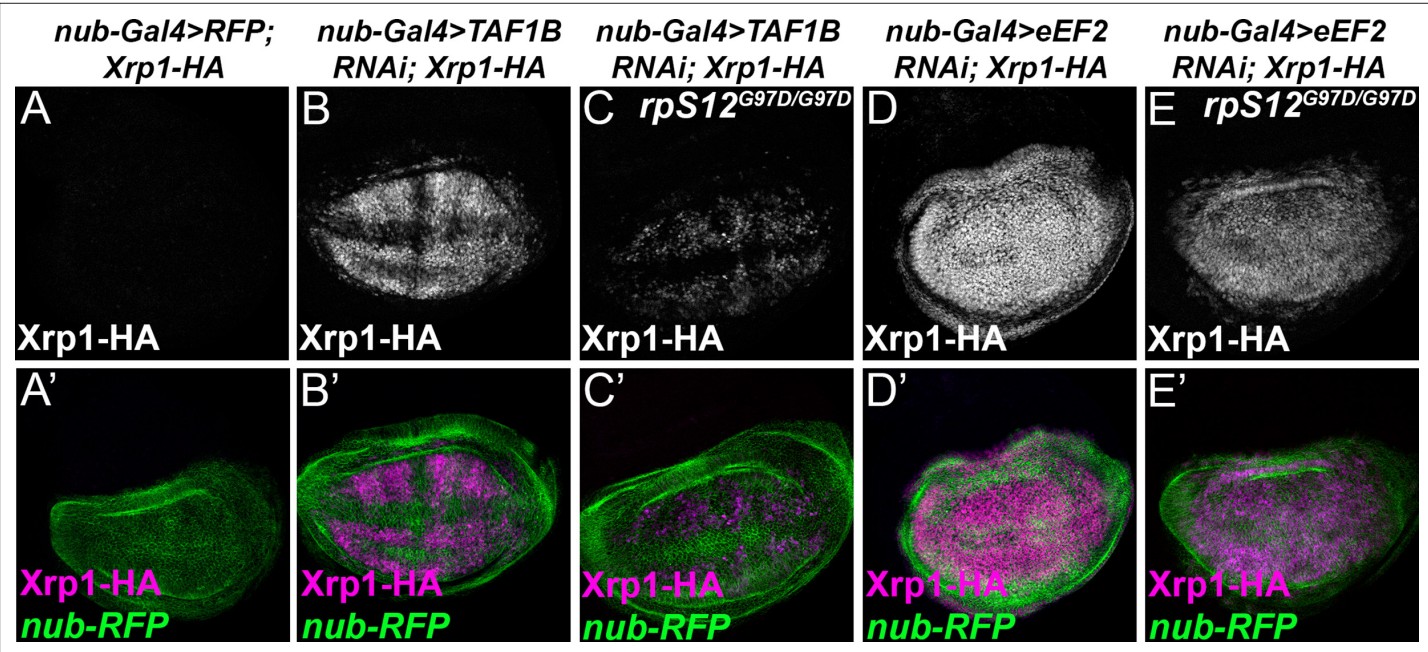

**Figure 9.** RpS12-dependence of Xrp1 expression. Figures show projections of Xrp1-HA expression from the wing discs of indicated genotypes. (**A**) Neglegible Xrp1-HA (magenta in A') was expressed in control discs where *nub-Gal4* drove only reporter RFP expression in the wing pouch (green in A'-E'). (**B**) TAF1B knockdown resulted in Xrp1-HA expression (magenta in B'). (**C**) Xrp1-HA expression was greatly reduced when TAF-1B was knocked-down in the *rpS12*$^{G97D}$ background (see also magenta in C'). (**D**) eEF2 knockdown resulted in strong Xrp1-HA expression (magenta in D'). (**E**) Xrp1-HA expression was only moderately reduced when eEF2 was knocked-down in the *rpS12*$^{G97D}$ background (see also magenta in E'). Genotypes: A: nubGal4, UAS-RFP/+; Xrp1-HA/Xrp1-HA, B: nubGal4, UAS-RFP/ UAS – RNAi$^{TAF1B}$;Xrp1-HA/ Xrp1-HA (line: v105873), C: nubGal4, UAS-RFP/ UAS – RNAi$^{TAF1B}$; *Rps12*$^{G97D}$, Xrp1-HA/ *Rps12*$^{G97D}$, Xrp1-HA, D: nubGal4, UAS-RFP/ UAS – RNAi$^{eEF2}$;Xrp1-HA/ Xrp1-HA, E: nubGal4, UAS-RFP/ UAS –RNAi$^{eEF2}$; *Rps12*$^{G97D}$, Xrp1-HA/ *Rps12*$^{G97D}$, Xrp1-HA.

completely prevented by *rpS12^G97D^*, a mis-sense allele that specifically affects this aspect of RpS12 function (*Lee et al., 2018*; *Ji et al., 2019*). We found that *rpS12^G97D^* homozygosity also reduced Xrp1 induction when TAF1B was depleted (*Figure 9A–C*), but had much less effect when eEF2 was depleted (*Figure 9D–E*). Thus, the mechanism of Xrp1 activation may resemble that in *Rp^+/-^* cells when rRNA synthesis is affected, but appears distinct when translation factors are inhibited.

## Xrp1 is a transcription factor that regulates cell competition

Xrp1 is a key mediator of multiple defects in ribosome biogenesis or function. Xrp1 is a sequence-specific DNA-binding protein implicated in genome maintenance, and binds directly to sequences of the P element whose transposition it promotes (*Akdemir et al., 2007*; *Francis et al., 2016*). Xrp1 also controls expression of many genes at the mRNA level (*Lee et al., 2018*), and other similar bZip proteins are transcription factors (*Tsukada et al., 2011*).

To test whether Xrp1 is a transcription factor, we used a dual-luciferase reporter system in transfected S2 cells (*Figure 10A–D*; *Figure 10—figure supplement 1*). Luciferase reporter plasmids were either based on the widely-used core promoter of the *Drosophila* hsp70 gene, or on a 400 bp genomic sequence spanning the transcription start site of the *Xrp1* gene itself (*Figure 10—figure supplement 2*). We cloned 8 x repeats of either of two different matches to the 10 bp Xrp1/Irbp18 consensus binding site in vitro (*Zhu et al., 2011*), which is similar to that recently deduced from ChIP-Seq following Xrp1 overexpression in vivo (*Baillon et al., 2018*) (Target 1 and Target 3) or of the sequence footprinted by Xrp1/Irbp18 on the P element terminal repeat (*Francis et al., 2016*) (target 2), which also contains a consensus match (*Figure 10A and B*). When Xrp1 expression was induced in transfected S2 cells, each of these Xrp1-binding sequences conferred 3x-8x activation of luciferase expression, whereas scrambled sequences were inactive (*Figure 10C, D*, *Figure 10—figure supplement 1A, B*). In the case of target 1, several-fold further induction was achieved by co-transfection and induction of Irbp18 expression, culminating in 23 x stimulation of luciferase expression by repeats of the Target 1 sequence in conjunction with the hsp70 basal promoter (*Figure 10—figure supplement 1A*). Irbp18 alone was inactive in the absence of transfected Xrp1 (*Figure 10C and D*; *Figure 10—figure supplement 1A, B*). Thus, the Xrp1/Irbp18 heterodimer stimulated transcription through its cognate binding sequences.

It has been suggested that an oxidative stress response in *Rp^+/-^* cells leads to competition with wild type cells (*Kucinski et al., 2017*; *Baumgartner et al., 2021*). *Rp^+/-^* cells express GstD1 reporters, whose transcription is activated by Nrf2, the master regulator of oxidative stress responses (*Kucinski et al., 2017*). Because the genes expressed in *Rp^+/-^* cells are also enriched for Xrp1 binding motifs, some of these genes might be activated directly by Xrp1, including GstD1 (*Ji et al., 2019 Figure 6—figure supplement 2*). The GstD1-GFP reporter used to report oxidative stress in *Rp^+/-^* cells contains a 2.7 kb genomic fragment that contains an Antioxidant Response Element (ARE) bound by the Nrf2/MafS dimer at position 1450–1460 (*Figure 10E*). Deletion of this motif abolishes GstD1-GFP induction in response to oxidative stress (*Sykiotis and Bohmann, 2008*). Recently, Brown et al identified Xrp1 binding motifs within the same GstD1-GFP reporter, and showed that these sequences are required for Xrp1-dependent induction in response to ER stress (*Brown et al., 2021*). We therefore compared induction of GstD1-GFP reporters in *Rp^+/-^* wing discs where the reporter sequences were either wild type, deleted for the Nrf2 binding motif, or mutated at the Xrp1-binding motifs (*Figure 10E*). We found that the Nrf2 binding motif was dispensable for GstD1-GFP induction in *Rp^+/-^* wing discs, whereas the Xrp1 sites were required, consistent with induction of GstD1-GFP and perhaps other genes as direct transcriptional targets of Xrp1, not Nrf2 (*Figure 10F–O*).

In addition to single copy genes, repetitive elements can also be regulated by Xrp1, as is revealed by re-analysis of previously published mRNA-seq data (*Lee et al., 2018*; *Ji et al., 2019*, *Supplementary file 1*). Transcription of the retrotransposon copia, for example, was elevated in RpS17 and RpS3 in an Xrp1-dependent (and RpS12-dependent) manner (*Figure 10P*, *Figure 10—figure supplement 3*). Accordingly, the regulatory, untranslated leader region of copia contains 7 copies of a motif closely matching the Xrp1 binding consensus, including 2 in a 28 bp region of dyad symmetry that is deleted from variants with reduced expression (*Figure 10Q*; *Mount and Rubin, 1985*; *Sneddon and Flavell, 1989*; *Matyunina et al., 1996*; *McDonald et al., 1997*; *Wilson et al., 1998*).

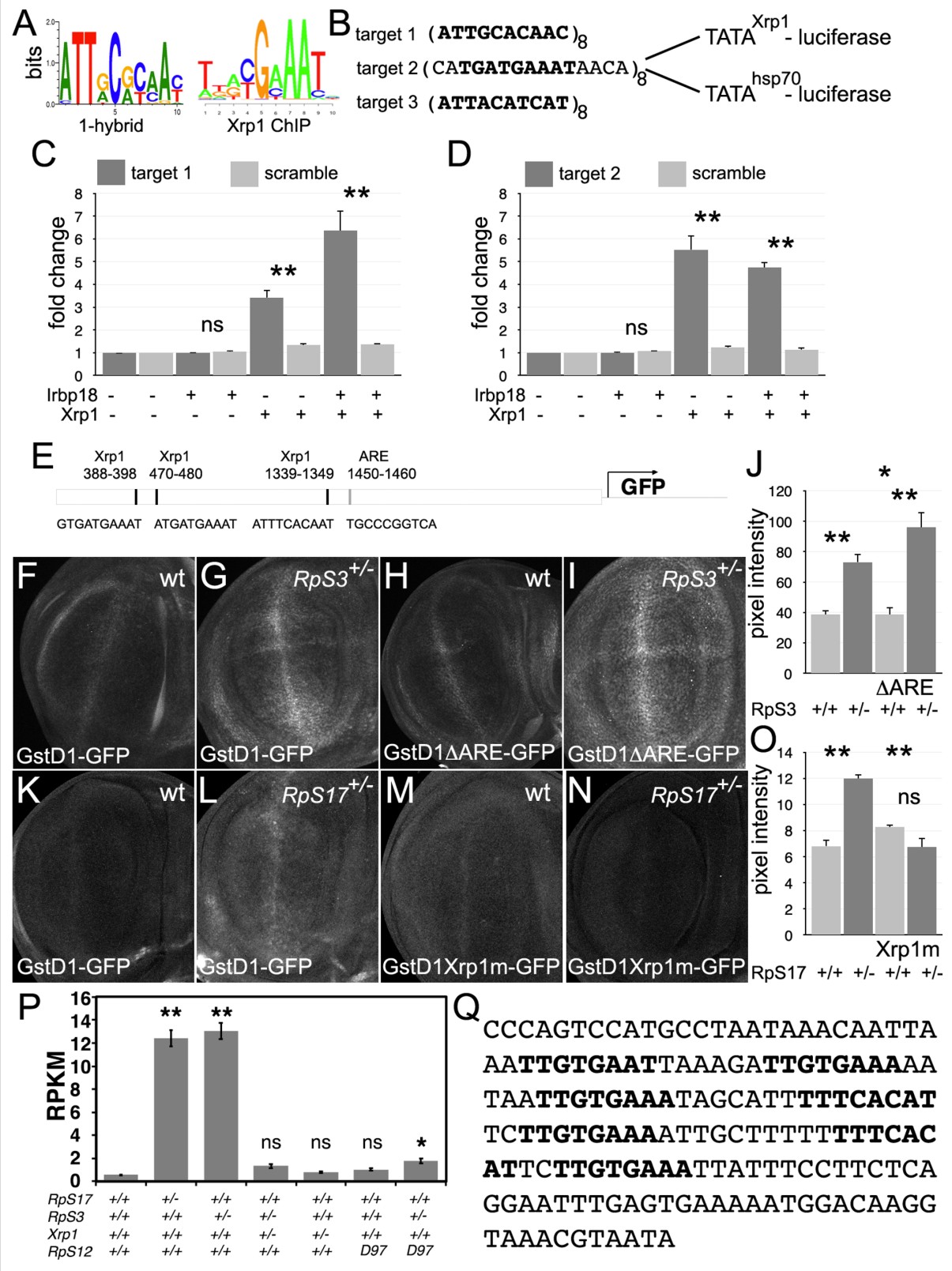

**Figure 10.** Transcriptional regulation by Xrp1. (**A**) Xrp1/Irbp18 binding consensus defined by bacterial 1-hybrid studies (***Zhu et al., 2011***) and by Xrp1 ChIP from *Drosophila* eye imaginal discs overexpressing an Xrp1-HA protein (***Baillon et al., 2018***). (**B**) Xrp1 binding motif sequences multimerized in luciferase reporter plasmids upstream of transcription start sites from the *Xrp1* gene or from the *hsp70* gene. Targets 1 and 3 were based on the 1-hybrid consensus, target 2 is the P element sequence footprinted by Xrp1-Irbp18 (***Francis et al., 2016***). The match to the consensus sites is shown

*Figure 10 continued on next page*

Figure 10 continued

in bold type. (**C**) Luciferase assays following transfection of reporters and protein expression plasmids into S2 cells. The target 1-TATA$^{Xrp1}$ reporter showed sequence-specific activation by transfected Xrp1. Transfected Irbp18 alone had no effect, but synergized with Xrp1. p-Values for comparisons between target one reporters and scrambled reporters were: Padj = 1, Padj = 0.00827, Padj = 3.47 × 10$^{-7}$, respectively. (**D**) Luciferase assays following transfection of reporters and protein expression plasmids into S2 cells. The target 2-TATA$^{Xrp1}$ reporter showed sequence-specific activation by transfected Xrp1. Transfected Irbp18 alone had no effect. p-Values for comparisons between target two reporters and scrambled reporters were: Padj = 1, Padj = 2.00 × 10$^{-8}$, Padj = 1.96 × 10$^{-7}$ respectively. (**E**) Potential regulatory sequences in the 2.7 kb upstream intergenic fragment used in the GstD1-GFP reporter (**Sykiotis and Bohmann, 2008**; **Brown et al., 2021**).3 Xrp1-binding motifs and the antioxidant response element (ARE) are indicated. (**F–I**) and (**K-N**) show projections from the central disc-proper regions of wing discs expressing reporter transgenes in the indicated genetic backgrounds. (**F**) Baseline GstD1-GFP expression in the wild-type wing disc. (**G**) Elevated GstD1-GFP expression in the $RpS3^{+/-}$ wing disc. (**H**) Baseline GstD1ΔARE-GFP expression in the wild-type wing disc. (**I**) Elevated GstD1ΔARE-GFP expression in the $RpS3^{+/-}$ wing disc. (**J**) Quantification of these results. Average pixel intensity from wing pouch regions was measured. Mean± SEM from multiple samples is shown. N = 5 for each genotype. Exact p values were: for GstD1-GFP in $RpS3^{+/-}$ compared to $RpS3^{+/+}$, Padj = 0.00257; for GstD1ΔARE-GFP in $RpS3^{+/-}$ compared to $RpS3^{+/+}$, Padj = 2.55 × 10$^{-5}$; for GstD1-GFP in $RpS3^{+/+}$ compared to GstD1ΔARE-GFP in $RpS3^{+/+}$, Padj = 0.993; for GstD1-GFP in $RpS3^{+/-}$ compared to GstD1ΔARE-GFP in $RpS3^{+/-}$, Padj = 0.0313. (**K**) baseline GstD1-GFP expression in the wild type wing disc. (**L**) Elevated GstD1-GFP expression in the $RpS17^{+/-}$ wing disc. (M) baseline expression of GstD1-GFP with all 3 Xrp1-binding motifs mutated in the wild type wing disc. (**N**) Expression of GstD1-GFP with all 3 Xrp1-binding motifs mutated was similar in the $RpS17^{+/-}$ wing disc to the wild type control. (**O**) Quantification of these results. Average pixel intensity from wing pouch regions was measured. Mean± SEM from multiple samples is shown. N = 5,6,5,6 for respective samples. Exact p values were: for GstD1-GFP in $RpS3^{+/-}$ compared to $RpS3^{+/+}$, Padj = 2.34 × 10$^{-6}$; for GstD1mXrp1-GFP in $RpS3^{+/-}$ compared to $RpS3^{+/+}$, Padj = 0.116; for GstD1-GFP in $RpS3^{+/+}$ compared to GstD1mXrp1-GFP in $RpS3^{+/+}$, Padj = 0.112; for GstD1-GFP in $RpS3^{+/-}$ compared to GstD1mXrp1-GFP in $RpS3^{+/-}$, Padj = 1.19 × 10$^{-6}$. (**P**) Pooled copia transcript levels for indicated genotypes determined from mRNA-seq data. Mean± standard deviation is shown. Values for individual copia insertions are shown in **Figure 10—figure supplement 2**. Asterisks indicate statistical significance of the difference from the wild type control: **, p < 0.01; *, p < 0.05; ns, p ≥ 0.05. Exact p values were: for RpS17$^{+/-}$ compared to wild type, p = 4 × 10$^{-14}$; for RpS3$^{+/-}$ compared to wild type, p = 2.33 × 10$^{-14}$; for RpS3$^{+/-}$ Xrp1$^{+/-}$ compared to wild type, p = 0.262; for RpS3$^{+/-}$ Xrp1$^{+/-}$ compared to wild type, p = 0.262;for Xrp1$^{+/-}$ compared to wild type, p = 0.494; for RpS3$^{+/-}$ Xrp1$^{+/-}$ compared to wild type, p = 0.262; for rpS12$^{D97/D97}$ compared to wild type, p = 0.858; for RpS3$^{+/-}$ rpS12$^{D97/D97}$ compared to wild type, p = 0.0201; for RpS3$^{+/-}$ rpS12$^{D97/D97}$ compared to wild type, p = 0.0201; for RpS3$^{+/-}$ Xrp1$^{+/-}$ compared to RpS3$^{+/-}$, p = 4.91 × 10$^{-14}$; for RpS3$^{+/-}$ Xrp1$^{+/-}$ compared to Xrp1$^{+/-}$, p = 0.635; for RpS3$^{+/-}$ rpS12$^{D97/D97}$ compared to rpS12$^{D97/D97}$, p = 0.251. Statistics:1-way ANOVA with Bonferroni-Holm correction for multiple testing was performed for the data shown in panels C,D,J,O,P. Data in panels C,D were based on triplicate measurements from each of three biological replicates for each transfection. Data in panel P were based on three biological replicates for each genotype. Genotypes: F: GstD1-GFP/+, G: GstD1-GFP/+; FRT82 $RpS3$ p{arm:LacZ}/+, H: GstD1ΔARE-GFP/+, I: GstD1ΔARE-GFP/+; FRT82 $RpS3$ p{arm:LacZ}/+, K: GstD1-GFP/+, L: GstD1-GFP; $RpS17$ p{arm:LacZ} FRT80B/+, M: GstD1 Xrp1m –GFP, N: GstD1Xrp1m-GFP; $RpS17$ p{arm:LacZ} FRT80B/+. Genotypes of P graph per column: 1$^{st}$: w$^{11-18}$; FRT82B/+, 2$^{nd}$: w$^{11-18}$; w p{hs:FLP}; $RpS17$ p{ubi:GFP} FRT80B/+,3$^{rd}$: w$^{11-18}$;w p{hs:FLP}; FRT82 $RpS3$ p{arm:LacZ/+, 4$^{th}$: w$^{11-18}$;w p{hs:FLP}; FRT82 $RpS3$ p{arm:LacZ/FRT82B FRT82B Xrp1$^{M2-73}$, 5$^{th}$: w$^{11-18}$; FRT82B Xrp1$^{M2-73}$ / +, 6$^{th}$: w$^{11-18}$; rpS12$^{D97}$ FRT80B / rpS12$^{D97}$ FRT80B, 7$^{th}$: w$^{11-18}$; rpS12$^{D97}$ FRT80B / rpS12$^{D97}$ FRT80B $RpS3$.

The online version of this article includes the following source data and figure supplement(s) for figure 10:

**Source data 1.** Luciferase data relevant to panels C,D.

**Source data 2.** GFP data relevant to panel J.

**Source data 3.** GFP data relevant to panel O.

**Source data 4.** mRNA-Seq data relevant to panel P, and also to **Figure 10—figure supplement 3**.

**Figure supplement 1.** Luciferase assays with *hsp70*-based reporter plasmids.

**Figure supplement 1—source data 1.** Luciferase measurements relevant to **Figure 10—figure supplement 1**.

**Figure supplement 2.** *Xrp1* promoter proximal sequences.

**Figure supplement 3.** Transcription of individual Copia elements.

## Discussion

We explored the mechanisms by which *Rp* mutations affect *Drosophila* imaginal disc cells, causing reduced translation and elimination by competition with wild-type cells in mosaics. Our findings reinforced the key role played by the AT-hook bZip protein Xrp1, which we showed is a sequence-specific transcription factor responsible for multiple aspects of not only the *Rp* phenotype, but also other ribosomal stresses (**Figure 11**). It was Xrp1, rather than the reduced levels of ribosomal subunits, that affected overall translation rate, primarily through PERK-dependent phosphorylation of eIF2α. Phosphorylation of eIF2α, as well as other disruptions to ribosome biogenesis and function such as reduction in rRNA synthesis or depletion of translation factors, were all sufficient to cause cell competition with nearby wild type cells, but this occurred because all these perturbations activated Xrp1, not because differences in translation levels between cells were sufficient to cause cell competition directly. In fact, our data show that differences in translation are neither sufficient nor necessary

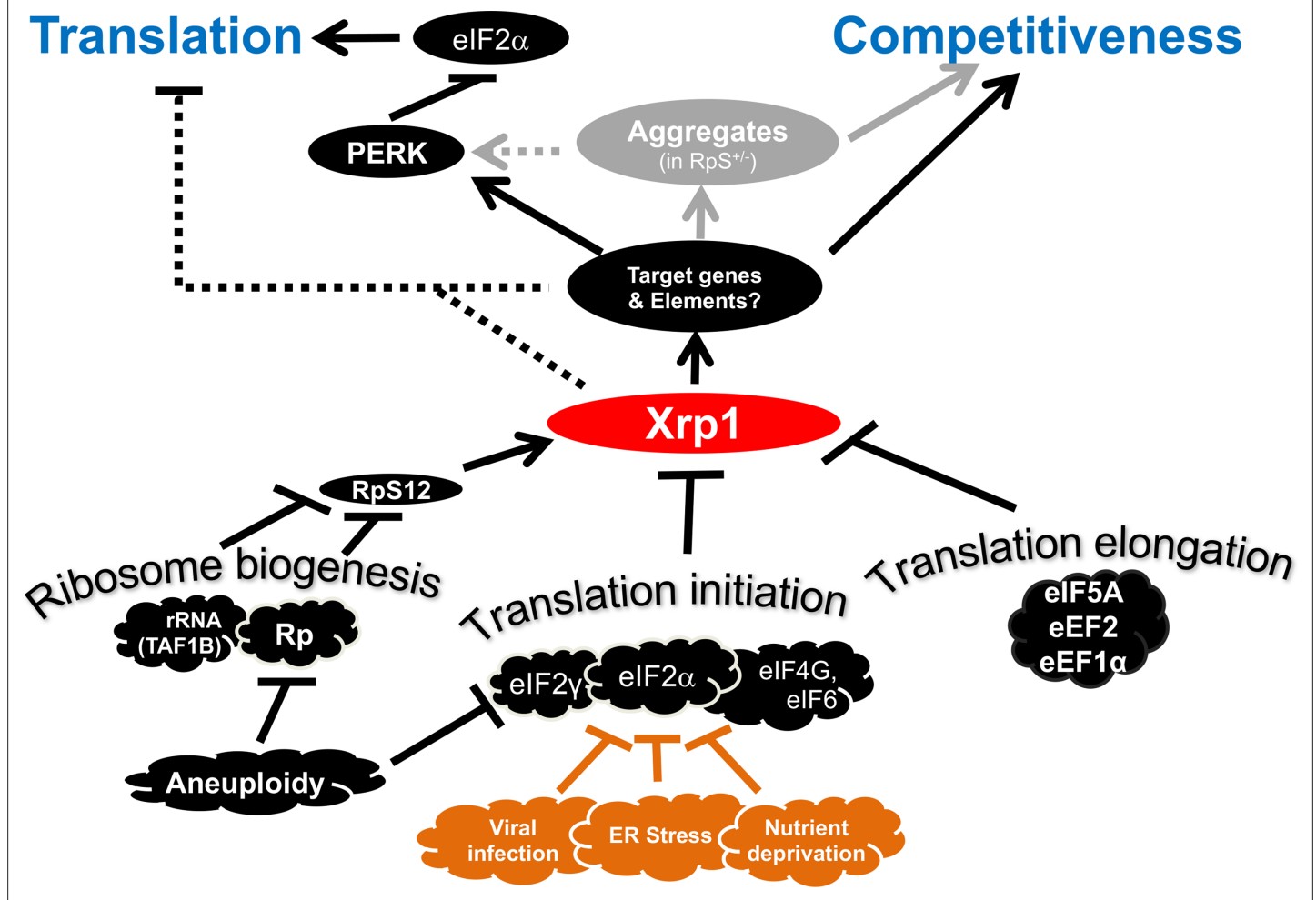

**Figure 11.** Transcriptional responses to Ribosome defects. Multiple consequences of defects in ribosome biogenesis, translation initiation, and translation elongation, depend on the transcription factor Xrp1 in the epithelial imaginal disc cells. Xrp1 is responsible for, or contributes to, reduced translation in response to these defects, through the PERK-dependent phosphorylation of eIF2α, a global regulator of CAP-dependent translation initiation. There is also evidence for some PERK-independent regulation of translation in genotypes such as TAF1B knock-down. Translation inhibition independently of Xrp1, which occurs after depletion of some translation factors, is not shown for simplicity. Xrp1 protein expression marks imaginal disc cells for elimination in competition with wild type cells. Differences in translation rate, including those caused by eIF2α phosphorylation or eIF2γ haploinsufficiency, are not sufficient to trigger cell competition without Xrp1. We speculate that other cellular stresses that phosphorylate eIF2α, including ER stress, nutrient deprivation, or (in mammals) infection with certain viruses might mark cells for competition, or interfere with cell competition that recognizes aneuploid cells on the basis of *Rp* or eIF2γ gene haploinsufficiency. It is notable that defective Tor signaling, which also reduces global translation rate, does not cause cell competition, (*Baumgartner et al., 2021*). Several pathways have been shown to induce Xrp1, including RpS12-dependent induction in *Rp*[+/-] cells and TAF1B-depleted cells (*Akdemir et al., 2007*; *Chapin et al., 2014*; *Lee et al., 2018*; *Ji et al., 2019*).

to trigger cell competition, which therefore depends on other Xrp1-dependent processes. Protein aggregation and activation of 'oxidative stress response' genes were also downstream effects of Xrp1 activity. While this paper was in preparation, other groups have also reported relationships between eIF2α phosphorylation, cell competition, and Xrp1 (*Baumgartner et al., 2021*; *Brown et al., 2021*; *Langton et al., 2021*; *Ochi et al., 2021*; *Recasens-Alvarez et al., 2021*),but none have reached the same overall conclusions as this study.

Our findings lead to a picture of Xrp1 as the key instigator of cell competition in response to multiple genetic triggers. Failure to appreciate the role of Xrp1 may have led to questionable conclusions in some previous studies. Our findings confirm the central importance of the transcriptional response to *Rp* mutations, and to other disruptions of ribosome biogenesis and function. They suggest therapeutic approaches to ribosomopathies, and have implications for the surveillance of aneuploid cells.

## Xrp1 activation by *Rp* gene haploinsufficiency

Rp gene haploinsufficiency has been proposed to affect ribosome concentrations, and hence translation, lead to the accumulation of ribosome components and assembly intermediates, and cause proteotoxic stress. Any of these could have been responsible for activating Xrp1 in $Rp^{+/-}$ cells.

Our data show that in fact ribosome subunit concentration is only moderately affected by *Rp* haploinsufficiency. We have seen 15–20% reduction in LSU concentrations in several *RpL* mutants, and 20–25% reduction in SSU concentrations in several *RpS* mutants. $RpL14^{+/-}$ also reduced SSU ~ 25%. Ribosomal subunit levels were unaffected by *Xrp1*. Broadly similar results have been reported in yeast (*Cheng et al., 2019*), and by mass spec quantification of ribosomal proteins in $RpS3^{+/-}$ and $RpS23^{+/-}$ *Drosophila* wing discs (*Baumgartner et al., 2021*; *Recasens-Alvarez et al., 2021*).

Multiple explanations for the modest effects on ribosome subunit number are possible. We particularly point out that, even if expression of a particular Rp is reduced in proportion to a 50% reduction in mRNA level, the respective protein concentration (i.e. number of molecules/cell volume) is unlikely to fall to 50%, because ribosomes are required for cellular growth, so that an *Rp* mutation affects the denominator in the concentration equation, as well as the numerator. It is even possible that a 50% reduction in its rate of Rp synthesis could leave steady state ribosome subunit concentration unaffected, if cellular growth rate was slowed by the same amount.

Modest changes in SSU and LSU levels could still affect ribosome function, which may depend more on the concentrations of free subunits than on total subunits. The data suggests, however, that cellular and animal models of DBA that have generally sought to achieve a 50% reduction in Rp protein expression (*Heijnen et al., 2014*; *Khajuria et al., 2018*) could be significantly more severe than occurs in DBA patients, and that actual ribosome subunit concentrations should be measured in DBA patient cells to guide future models.

We confirmed that ribosome assembly intermediates accumulate in *Drosophila* wing discs following *Rp* haploinsufficiency. In yeast, aggregates of unused Rp rapidly trigger transcriptional changes (*Albert et al., 2019*; *Tye et al., 2019*). It has been suggested proteotoxic stress might lead to eIF2α phosphorylation in *Drosophila* (*Baumgartner et al., 2021*; *Recasens-Alvarez et al., 2021*), with Xrp1 amplifying this effect (*Langton et al., 2021*), but we found that while Perk was responsible for eIF2α phosphorylation, it was not required for Xrp1 expression in *Rp* mutants, placing Perk and eIF2α phosphorylation downstream. Consistent with this, we show that the protein aggregates reported in $Rp^{+/-}$ cells (*Baumgartner et al., 2021*; *Recasens-Alvarez et al., 2021*) were only seen in some Rp mutants, all affecting the SSU, and were also a downstream consequence of Xrp1 activity, as also now seen by others (*Langton et al., 2021*). It remains plausible that unused ribosomal components are the initial trigger for cellular responses in *Drosophila* as in yeast, but in *Drosophila* the species involved have not yet been identified. Because Xrp1 expression depends particularly on RpS12, an RpS12-containing signaling species is one possibility (*Kale et al., 2018*; *Lee et al., 2018*; *Boulan et al., 2019*; *Ji et al., 2019*).

## *Rp* mutants affect global translation rate through eIF2α

PERK-dependent phosphorylation of eIF2α was the mechanism by which Xrp1 suppresses global translation in $Rp^{+/-}$ mutants.

It is interesting that Xrp1 protein levels increase under conditions of reduced global translation. Perhaps Xrp1 is one of the few genes whose translation is enhanced when eIF2α is phosphorylated (*Wek, 2018*; *Brown et al., 2021*). Although PERK is known to be activated by ER stress, the IRE/Xbp1 branch of the UPR was not unequivocally detected in $Rp^{+/-}$ mutants. We suspect that the UPR might be suppressed in $Rp^{+/-}$ mutants by Xrp1-dependent changes in transcription of Perk, BiP, and other UPR genes (*Figure 11*). Perhaps in proliferative tissues it is preferable to replace stressed cells than to repair them.

It will be interesting to determine whether eIF2α phosphorylation occurs in human ribosomopathies. Notably, knock-out of CReP, one of the two mouse PPP1R15 homologs, causes anemia, similar to DBA (*Harding et al., 2009*; *Da Costa et al., 2018*), and PERK-dependent eIF2α phosphorylation occurs in RpL22-deficient mouse αβ T-cells and activates p53 there (*Solanki et al., 2016*). Thus, inhibitors of eIF2α phosphorylation could be explored as potential DBA drugs. TAF1B depletion, which also acted through Xrp1 and eIF2α phosphorylation in *Drosophila*, is a model of Treacher Collins Syndrome (*Trainor et al., 2008*), and failure to release eIF6, leading to defective LSU maturation and

80 S ribosome formation, causes Schwachman Diamond syndrome (*Warren, 2018*), two other ribosomopathies where potential contributions of eIF2α phosphorylation are possible.

## Xrp1, not differential translation, causes competition between cells

Because eIF2α phosphorylation alone was sufficient to target cells for competitive elimination, at first it seemed that eIF2α phosphorylation was the mechanism by which Xrp1 caused cell competition, which often correlates with differences in cellular translation levels (*Nagata et al., 2019*). One group has suggested this (*Ochi et al., 2021*). Another group concluded that eIF2α phosphorylation in *Rp*⁺/⁻ cells did not lead to cell competition (*Baumgartner et al., 2021*), but the opposite conclusion is corroborated by the independent finding that haploinsufficiency for the γ subunit of eIF2 also causes cell competition (*Ji et al., 2021*). Our conclusion is that eIF2α phosphorylation can cause cell competition but not directly. Instead, phosphorylation of eIF2α is itself sufficient to activate Xrp1 expression, as found by us and by several other groups (*Brown et al., 2021*; *Langton et al., 2021*; *Ochi et al., 2021*). Crucially, *Perk* inactivation restored eIF2α phosphorylation and global translation to normal in *Rp*⁺/⁻ cells (*Figure 3H, I*, *Figure 3—figure supplement 1G-L*), without preventing cell competition, which must therefore depend on other Xrp1 targets (*Figure 5—figure supplement 3*). Elimination of *eIF2γ* haploinsufficient cells is also Xrp1-dependent, as expected if Xrp1 is downstream of eIF2 activity in cell competition (*Ji et al., 2021*).

Knock-down of factors directly involved in the translation mechanism further distinguished cell competition from differential translation levels. Different factors affected translation in diverse ways. In *Rp*⁺/⁻ mutants, PERK-dependent phosphorylation of eIF2α suppressed global translation, which was normalized by Perk or Xrp1 depletion. PERK-dependent phosphorylation of eIF2α also contributed to the translation deficits of cells depleted for TAF1B, eIF6, and possibly eEF1α1, which were all partially restored by eIF2α dephosphorylation and fully by Xrp1 depletion, suggesting that Xrp1 can also affect translation by additional mechanisms. By contrast, translation deficits caused by eIF4G, eIF5A, or eEF2 depletion were restored little by eIF2α dephosphorylation or Xrp1 depletion, indicating Xrp1-independent effects of these factors on translation.

Several conclusions follow from studies of these factors. As noted above, reduced translation cannot be required for cell competition, because *perk*⁻/⁻ *Rp*⁺/⁻ mutant cells are eliminated by *perk*⁺/⁻ *Rp*⁺/⁺ cells (*Figure 5—figure supplement 3*). Secondly, lower translation is not sufficient for competitive elimination, because no competitive cell death was observed in eIF4G Xrp1-depleted, eIF5A Xrp1-depleted, and eEF2 Xrp1-depleted cells, even though their translation was lower than the nearby wild type cells. Another group also concluded that lower translation alone was not sufficient for cell competition, based on different data (*Baumgartner et al., 2021*).

Our findings focus attention on Xrp1 activity as the key factor marking cells for competition, distinct from its effects on global translation, which only trigger cell competition when Xrp1 is induced (*Figure 11*).

## Transcriptional regulation of cell competition

We confirm that Xrp1 is a sequence-specific transcriptional activator, and propose that direct transcriptional targets of Xrp1 predispose *Rp*⁺/⁻ cells, and other genotypes, to elimination by wild-type cells (*Figure 11*). Expression of several hundred single copy genes is regulated by Xrp1 in Rp mutant cells, and we report here that expression of some transposable elements is affected in addition, whose potential contribution to cell competition might also be interesting (*Figure 10P*, *Figure 10—figure supplement 3*, *Supplementary file 1*). One or more of these transcriptional targets may lead to competitive interactions with wild-type cells.

These Xrp1 targets include genes that also contribute to oxidative stress responses, such as GstD genes, which has previously led to the suggestion that an oxidative stress response is responsible for cell competition (*Kucinski et al., 2017*; *Baumgartner et al., 2021*; *Recasens-Alvarez et al., 2021*). Because the oxidative stress reporter used in previous studies is probably activated in *Rp*⁺/⁻ cells by direct Xrp1-binding, and not by the Nrf2-dependent ARE site, it is not now certain whether *Rp*⁺/⁻ cells experience oxidative stress or Nrf2 activity (*Figure 10*). An alternative explanation of cell competition in response to Nrf2 over-expression (*Kucinski et al., 2017*) could be induction of Xrp1 expression by Nrf2 (*Langton et al., 2021*).

## Xrp1 as a central orchestrator of cell competition

Our results reveal the central importance of Xrp1 as the driver of cell competition (*Figure 11*). Far from being expressed specifically in *Rp* mutants, we now find that Xrp1 is induced by multiple challenges, not only to ribosome biogenesis, such as by depletion of the polI cofactor TAF1B or LSU maturation factor eIF6, but also challenges to ribosome function, both at the levels of initiation or elongation, all leading to cell competition and to Xrp1-dependent eIF2α phosphorylation (*Figure 11*).

Had we not evaluated Xrp1 expression and function in PPP1R15-depleted cells, we would have concluded that eIF2α phosphorylation was the likely downstream effector of competition in *Rp* mutant cells, rather than an example of another upstream stress that induces Xrp1 (*Figure 11*). It is becoming apparent that other triggers of cell competition, including depletion for Helicase at 25E (Hel25E), a helicase that plays roles in mRNA splicing and in mRNA nuclear export, over-expression of Nrf2, the transcriptional master regulator of the oxidative stress response, and loss of mahjong, a ubiquitin ligase implicated in planar cell polarity, all lead to Xrp1 expression (*Langton et al., 2021*; *Ochi et al., 2021*; Kumar and Baker,unpublished). Earlier models regarding these cell competition mechanisms, in which the role of Xrp1 was not recognized, may be questionable. It would be important now to check for possible activation of Xrp1 in cells with other defects affecting translation, including mutations of an eIF5A-modifying enzyme (*Patel et al., 2009*) and mutations of a pre-rRNA processing enzyme (*Zielke et al., 2020*). It would not be surprising if other conditions that lead to eIF2α phosphorylation, such as ER stress, nutrient deprivation, or viral infection (*Ron and Walter, 2007*; *Hetz, 2012*), also activate Xrp1 and are thereby marked for elimination by more normal neighbors (*Figure 11*). It will be particularly interesting to determine whether any of these environmental perturbations could interfere with surveillance and removal of aneuploid cells, given the potential importance for tumor surveillance (*Ji et al., 2021*).

# Materials and methods

**Key resources table**

| Reagent type (species) or resource | Designation | Source or reference | Identifiers | Additional information |
|---|---|---|---|---|
| Gene (*Drosophila melanogaster*) | Xrp1 | GenBank | FLYBASE: FBgn0261113 | |
| Gene (*Drosophila melanogaster*) | RpS12 | GenBank | FLYBASE: FBgn0286213 | |
| Gene (*Drosophila melanogaster*) | RpS18 | GenBank | FLYBASE: FBgn0010411 | |
| Gene (*Drosophila melanogaster*) | RpL27A | GenBank | FLYBASE: FBgn0285948 | |
| Gene (*Drosophila melanogaster*) | RpS3 | GenBank | FLYBASE: FBgn0002622 | |
| Gene (*Drosophila melanogaster*) | RpS17 | GenBank | FLYBASE: FBgn0005533 | |
| Gene (*Drosophila melanogaster*) | RpL14 | GenBank | FLYBASE: FBgn0017579 | |
| Gene (*Drosophila melanogaster*) | RpL19 | GenBank | FLYBASE: FBgn0285950 | |
| Gene (*Drosophila melanogaster*) | RpL36 | GenBank | FLYBASE: FBgn0002579 | |
| Gene (*Drosophila melanogaster*) | TAF1B | GenBank | FLYBASE: FBgn0037792 | |
| Gene (*Drosophila melanogaster*) | PPP1R15 | GenBank | FLYBASE: FBgn0034948 | |
| Gene (*Drosophila melanogaster*) | PERK | GenBank | FLYBASE: FBgn0037327 | |

*Continued on next page*

*Continued*

| Reagent type (species) or resource | Designation | Source or reference | Identifiers | Additional information |
|---|---|---|---|---|
| Gene (*Drosophila melanogaster*) | Gcn2 | GenBank | FLYBASE: FBgn0019990 | |
| Gene (*Drosophila melanogaster*) | Irbp18 | GenBank | FLYBASE: FBgn0036126 | |
| Gene (*Drosophila melanogaster*) | eIF4G | GenBank | FLYBASE: FBgn0023213 | |
| Gene (*Drosophila melanogaster*) | eEF2 | GenBank | FLYBASE: FBgn0000559 | |
| Gene (*Drosophila melanogaster*) | eIF6 | GenBank | FLYBASE: FBgn0034915 | |
| Gene (*Drosophila melanogaster*) | copia | GenBank | FLYBASE: FBgn0013437 | |
| Genetic reagent (*D. melanogaster*) | eEF1α1 | GenBank | FLYBASE: FBgn0284245 | |
| Genetic reagent (*D. melanogaster*) | eIF5A | GenBank | FLYBASE: FBgn0285952 | |
| Genetic reagent (*D. melanogaster*) | Xrp1[HA] | *Blanco et al., 2020* | | Strain maintained in Dr. Nicholas Baker's lab. |
| Genetic reagent (*D. melanogaster*) | Xrp1[M2-73] allele | *Lee et al., 2018* | FLYBASE: RRID:BDSC_81270 | Bloomington *Drosophila* Stock Center #81,270 |
| Genetic reagent (*D. melanogaster*) | RpS12[G97D] allele | *Tyler et al., 2007* | FLYBASE: FBal0193403 | Strain maintained in Dr. Nicholas Baker's lab. |
| Genetic reagent (*D. melanogaster*) | UAS-dsRNA[Xrp1] | *Perkins et al., 2015* | FLYBASE: RRID:BDSC_34521 | Bloomington *Drosophila* Stock Center #34,521 |
| Genetic reagent (*D. melanogaster*) | UAS-dsRNA[Xrp1] | *Dietzl et al., 2007* | FLYBASE: FBti0118620 | Vienna *Drosophila* Resource Center #v 107,860 |
| Genetic reagent (*D. melanogaster*) | UAS-dsRNA[irbp18] | *Perkins et al., 2015* | FLYBASE: RRID:BDSC_33652 | Bloomington *Drosophila* Stock Center #33,652 |
| Genetic reagent (*D. melanogaster*) | UAS-dsRNA[w] | *Perkins et al., 2015* | FLYBASE: RRID:BDSC_33623 | Bloomington *Drosophila* Stock Center #33,623 |
| Genetic reagent (*D. melanogaster*) | arm-LacZ | *Vincent et al., 1994* | FLYBASE: FBal0040819 | |
| Genetic reagent (*D. melanogaster*) | Ubi-GFP | *Davis et al., 1995* | FLYBASE: FBal0047085 | |
| Genetic reagent (*D. melanogaster*) | M(2)56 F (mutating RpS18) | Laboratory of Y. Hiromi | FLYBASE: FBal0011916 | |
| Genetic reagent (*D. melanogaster*) | Df(1)R194 (deleting RpL36) | *Duffy et al., 1996* | FLYBASE: FBab0024817 | |
| Genetic reagent (*D. melanogaster*) | P{RpL36+} | *Tyler et al., 2007* | FLYBASE: FBal0193398 | |
| Genetic reagent (*D. melanogaster*) | M{RpL19+} | *Baillon et al., 2018* | | |
| Genetic reagent (*D. melanogaster*) | Df(2 R)M60E (deleting RpL19) | *Baillon et al., 2018* | FLYBASE: FBab0001997 | |

*Continued on next page*

*Continued*

| Reagent type (species) or resource | Designation | Source or reference | Identifiers | Additional information |
|---|---|---|---|---|
| Genetic reagent (*D. melanogaster*) | hs-FLP | *Struhl and Basler, 1993* | FLYBASE: FBtp0001101 | |
| Genetic reagent (*D. melanogaster*) | P{GAL4-Act5C(FRT.CD2).P}S | *Pignoni and Zipursky, 1997* | FLYBASE: FBti0012408 | Bloomington *Drosophila* Stock Center #51,308 |
| Genetic reagent (*D. melanogaster*) | P{neoFRT}42D | *Xu and Rubin, 1993* | FLYBASE: FBti0141188 | Bloomington *Drosophila* Stock Center #1,802 |
| Genetic reagent (*D. melanogaster*) | P{neoFRT}80B | *Xu and Rubin, 1993* | FLYBASE: FBti0002073 | Bloomington *Drosophila* Stock Center #1988 |
| Genetic reagent (*D. melanogaster*) | P{neoFRT}82B | This study | FLYBASE: FBti0002074 | Viable line derived from Bloomington Drosophila Stock Center lines BL5188 and BL30555 |
| Genetic reagent (*D. melanogaster*) | UAS-dsRNA$^{TAF1B}$ | *Perkins et al., 2015* | RRID:BDSC_61957 | Bloomington *Drosophila* Stock Center #61,957 |
| Genetic reagent (*D. melanogaster*) | UAS-dsRNA$^{TAF1B}$ | *Dietzl et al., 2007* | FLYBASE: FBti0118760 | Vienna *Drosophila* Resource Center #v105873 |
| Genetic reagent (*D. melanogaster*) | UAS-dsRNA$^{eIF6}$ | *Dietzl et al., 2007* | FLYBASE: FBti0116845 | Vienna *Drosophila* Resource Center #v108094 |
| Genetic reagent (*D. melanogaster*) | UAS-dsRNA$^{eIF4G}$ | *Dietzl et al., 2007* | FLYBASE: FBti0095456 | Vienna *Drosophila* Resource Center #v17002 |
| Genetic reagent (*D. melanogaster*) | UAS-dsRNA$^{eIF5A}$ | *Dietzl et al., 2007* | FLYBASE: FBti0121478 | Vienna *Drosophila* Resource Center #v101513 |
| Genetic reagent (*D. melanogaster*) | UAS-dsRNA$^{eEF2}$ | *Dietzl et al., 2007* | FLYBASE: FBti0117284 | Vienna *Drosophila* Resource Center #v107268 |
| Genetic reagent (*D. melanogaster*) | UAS-dsRNA$^{eEF1\alpha1}$ | *Dietzl et al., 2007* | FLYBASE: FBti0121842 | Vienna *Drosophila* Resource Center #v104502 |
| Genetic reagent (*D. melanogaster*) | UAS-dsRNA$^{PERK}$ | *Dietzl et al., 2007* | FLYBASE: FBti0141304 | Vienna *Drosophila* Resource Center # v110278 |
| Genetic reagent (*D. melanogaster*) | UAS-dsRNA$^{PERK}$ | *Dietzl et al., 2007* | FLYBASE: FBti0093363 | Vienna *Drosophila* Resource Center #v 16,427 |
| Genetic reagent (*D. melanogaster*) | UAS-dsRNA$^{Gcn2}$ | *Dietzl et al., 2007* | FLYBASE: FBti0118018 | Vienna *Drosophila* Resource Center #v103976 |
| Genetic reagent (*D. melanogaster*) | *RpS18* mutation *M(2 R)56 f* | Laboratory of Y. Hiromi | FLYBASE: FBal0284387 | |
| Genetic reagent (*D. melanogaster*) | RpS3 | *Burke and Basler, 1996* | Flybase: FBgn0002622 | |
| Genetic reagent (*D. melanogaster*) | *RpL27A* Df(2 L) M24F11 | *Marygold et al., 2007* | Flybase: FBab0001492 | |
| Genetic reagent (*D. melanogaster*) | RpS17 mutation M(3 L)67 C$^4$ | *Morata and Ripoll, 1975* | Flybase: FBal0011935 | |
| Genetic reagent (*D. melanogaster*) | en-Gal4 | *Neufeld et al., 1998* | RRID:BDSC_6356 | |
| Genetic reagent (*D. melanogaster*) | UAS-S65T-GFP | FBrf0086268 | FBtp0001403 | |
| Genetic reagent (*D. melanogaster*) | P[GAL4-Act5C(FRT.CD2).P]S | FBrf0221941 | FBti0012408 | |

*Continued*

| Reagent type (species) or resource | Designation | Source or reference | Identifiers | Additional information |
|---|---|---|---|---|
| Genetic reagent (*D. melanogaster*) | P[UAS-His-RFP]3 | FBrf0221941 | FBti0152909 | |
| Antibody | anti-active-Dcp1 (rabbit polyclonal) | Cell Signalling Technology | Cat #9,578 RRID:AB_2721060 | (1:100) |
| Antibody | anti-XRP1(short) (rabbit polyclonal) | *Francis et al., 2016* | | (1:200) |
| Antibody | antiphospho-RpS6 (rabbit polyclonal) | *Romero-Pozuelo et al., 2017* | | (1:200) |
| Antibody | anti-p62 (rabbit polyclonal) | *Pircs et al., 2012* | | (1:300) |
| Antibody | anti-phospho-eIF2α (rabbit polyclonal) | Thermo Fisher Scientific | Cat #44–728 G RRID:AB_2533736 | (1:200) |
| Antibody | anti-phospho-eIF2α (D9G8) (rabbit monoclonal) | Cell Signaling Technology | Cat #D9G8 #3398 RRID:AB_10829234 | (1:200) |
| Antibody | anti-HA Tag (mouse monoclonal) | Cell Signalling Technology | Cat #2,367 RRID:AB_10691311 | (1:100) |
| Antibody | anti-beta galactosidase (mAb40-1a) (mouse monoclonal) | DSHB | RRID: AB_2314509 | (1:100) |
| Antibody | anti-Mouse IgG, Cy2 (goat monoclonal) | Jackson Immunoreseach | Cat #115-225-166 RRID:AB_2338746 | (1:200) |
| Antibody | anti-Mouse IgG, Alexa Fluor 555 (goat polyclonal) | Thermo Fischer Scientific | Cat #A28180 RRID:AB_2536164 | (1:200) |
| Antibody | anti-Mouse IgG, Alexa Fluor 647 (Goat polyclonal) | Thermo Fischer Scientific | Cat #A-21235 RRID:AB_2535804 | (1:200) |
| Antibody | anti-Mouse IgG, Alexa Fluor 488 (Goat polyclonal) | Thermo Fischer Scientific | Cat #A-11001 RRID:AB_2534069 | (1:400) |
| Antibody | anti-Rabbit Cy3, (Goat polyclonal) | Thermo Fischer Scientific | Cat #A-21244 RRID:AB_2535812 | (1:200) |

*Continued*

| Reagent type (species) or resource | Designation | Source or reference | Identifiers | Additional information |
|---|---|---|---|---|
| Antibody | anti-Rabbit IgG, Alexa Fluor 555 (Goat polyclonal) | Thermo Fischer Scientific | Cat #A21429 RRID:AB_2535850 | (1:300) |
| Antibody | anti-Rabbit IgG, Alexa Fluor 647 (Goat polyclonal) | Thermo Fischer Scientific | Cat #A-21244 RRID:AB_2535812 | (1:200) |
| Antibody | anti-Guinea Pig Cy5 (Donkey polyclonal) | Jackson Immunoresearch | Cat #706-175-148 RRID:AB_2340462 | (1:200) |
| Antibody | Anti-rRNA (mouse monoclonal Y10b) | Thermo Fisher Scientific *Lerner et al., 1981* | Cat #MA1-13017 RRID:AB_10979967 | (1:100) |
| Antibody | anti-dRpS12 (guinea-pig polyclonal) | *Kale et al., 2018* | | (1:100) |
| Antibody | rabbit anti-hRpL10Ab (rabbit polyclonal) | Sigma-Aldrich | Cat #SAB1101199; RRID: AB_10620774 | (1:200) |
| Antibody | anti-Rack1 (rabbit monoclonal) | Cell Signalling Technology | Cat #D59D5 RRID:AB_10705522 | (1:100) |
| Antibody | anti-RpS9 (rabbit monoclonal) | Abcam | Cat #ab117861 RRID:AB_10933850 | (1:100) |
| Antibody | Anti-RpL9 (rabbit monoclonal) | Abcam | Cat #ab50384 RRID:AB_882391 | (1:100) |
| commercial assay, kit | Maxiscript T7 Transcription kit | Ambion | Cat #AM1312 | |
| other | ULTRAhyb-Oligo buffer | Ambion | Cat #AM8663 | |
| commercial assay, kit | Click-iT Plus OPP Alexa Fluor 594 or 488 Protein Synthesis Assay Kit | Thermo Fisher Scientific | Cat #C10457 | |
| Chemical compound, drug | Biotin-16-UTP | Roche | Cat #11388908910 | |
| Chemical compound, drug | RNA Sample Loading Buffer | Sigma-Aldrich | Cat #R4268-5VL | |
| Chemical compound, drug | Heat inactivated Fetal Bovine Serum | Gibco | Cat #10082139 | |

*Continued on next page*

*Continued*

| Reagent type (species) or resource | Designation | Source or reference | Identifiers | Additional information |
|---|---|---|---|---|
| Chemical compound, drug | Schneider's *Drosophila* Medium | Gibco | Cat #21720024 | |
| Chemical compound, drug | Trizol | Ambion | Cat #15596–026 | |
| Chemical compound, drug | Odyssey Blocking buffer (PBS) | Li-COR | Cat #927–40003 | |
| sequence-based reagent | 18 S probe_Forward | *Lee et al., 2018* | Invitrogen | GGTGCTGAAGCTTATGTAGC |
| sequence-based reagent | 18 S probe_Reverse | *Lee et al., 2018* | Invitrogen | TAATACGACTCACTATAGGGAGACAAAGGGCA GGGACG |
| sequence-based reagent | 5.8 S probe_Forward | *Lee et al., 2018* | Invitrogen | GCTTATATGAAACTAAGACATTTCG |
| sequence-based reagent | 5.8 S probe_Reverse | *Lee et al., 2018* | Invitrogen | TAATACGACTCACTATAGGGTACATAAC AGCAT GGACTGC |
| sequence-based reagent | ITS2 probe_Forward | This study | Invitrogen | 5'- CTTTAATTAATTTTATAGTGCTGCTTGG-3' |
| sequence-based reagent | ITS2 probe_reverse | This study | Invitrogen | 5'- TAATACGACTCACTATAGGGTTGT ATATAACTTTATCTTG-3' |
| sequence-based reagent | 28 S probe_Forward | This study | Invitrogen | 5'-GCAGAGAGATATGGTAGATGGGC –3' |
| sequence-based reagent | 28 S probe_reverse | This study | Invitrogen | 5'- TAATACGACTCACTATAGGGTTCCAC AATTGGCTACGTAACT-3' |
| sequence-based reagent | ITS1 probe_Forward | This study | Invitrogen | 5'- GGAAGGATCATTATTGTATAATATC-3' |
| sequence-based reagent | ITS1 probe_Reverse | This study | Invitrogen | 5'- TAATACGACTCACTATAGGGATG ATTACCACACATTCG-3' |
| sequence-based reagent | 7SL probe_Forward | This study | Invitrogen | 5'- TCGACTGGAAGGTTGGCAGCTTCTG-3' |
| sequence-based reagent | 7SL probe_Reverse | This study | Invitrogen | 5'- TAATACGACTCACTATAGGGATTGTGG TCCAACCATATCG-3' |
| Other | VECTASHIELD antifade mounting medium | Vector Laboratories | Cat #H-1000 | |
| Other | Nuclear Mask reagent | Thermo Fisher Scientific | Cat #H10325 | |
| Cell line | S2-DGRC | *Drosophila* Genomics Resource Center (NIH Grant 2P40OD010949) | FLYBASE: FBtc0000006 RRID:CVCL_TZ72 | Stock #6 (*D. melanogaster* embryonic cell line) |
| Other | Dual-Luciferase Reporter Assay System | Promega | Cat #E1910 | |
| Other | TransIT-2020 Transfection Reagent | Mirus Bio | Cat #MIR 5404 | |
| sequence-based reagent | Xrp target 1+ strand | This study | | TCGAGATTGCACAACGCTCATTGCAC AACGTTCATTGCACAACGGCAATTGCACAACG |

*Continued on next page*

*Continued*

| Reagent type (species) or resource | Designation | Source or reference | Identifiers | Additional information |
|---|---|---|---|---|
| sequence-based reagent | Xrp target 1 - strand | This study | | TCGACGTTGTGCAATTGCCGTTGTGCAATG AACGTTGTGCAATGAGCGTTGTGCAATC |
| sequence-based reagent | Xrp target 2+ strand | This study | | TCGAGCATGATGAAATAACATGCTCCATGA TGAAATAACATGTTCCATGATGAAATAACA TGGCACATGATGAAATAACATG |
| sequence-based reagent | Xrp target 2 - strand | This study | | TCGACATGTTATTTCATCATGTGCCATGTTA TTTCATCATGGAACATGTTATTTCATCATG GAGCATGTTATTTCATCATGC |
| sequence-based reagent | Xrp target 3+ strand | This study | | TCGAGATTACATCATGCTCATTACATC ATGTTCATTACATCATGGCAATTACATCATG |
| sequence-based reagent | Xrp target 3 - strand | This study | | TCGACATGATGTAATTGCCATGATGTAATG AACATGATGTAATGAGCATGATGTAATC |
| sequence-based reagent | Xrp trgt 1+ strand shuffled | This study | | TCGAGTGACAACTCAGCTCTGACAACTCAGT TCTGACAACTCAGGCATGACAACTCAG |
| sequence-based reagent | Xrp trgt 1 - strand shuffled | This study | | TCGACTGAGTTGTCATGCCTGAGTTGTCAGAA CTGAGTTGTCAGAGCTGAGTTGTCAC |
| sequence-based reagent | Xrp trgt 2+ strand shuffled | This study | | TCGAGTTCAAATCAATAGGAAGCTCTTCAAATCA ATAGGAAGTTCTTCAAATCAATAGGAAGGCATTC AAATCAATAGGAAG |
| sequence-based reagent | Xrp trgt 2 - strand shuffled | This study | | TCGACTTCCTATTGATTTGAATGCCTTCCTATT GATTTGAAGAACTTCCTATTGATTTGAAGAGC TTCCTATTGATTTGAAC |
| recombinant DNA reagent | pGL3-Promoter Vector | Promega | Cat #E1761 | |
| recombinant DNA reagent | pAct5.1/V5-His C vector | Thermo Fischer Scientific | Cat #V411020 | |
| recombinant DNA reagent | pIS1 plasmid | Addgene | Cat #12,179 | |
| recombinant DNA reagent | pUAST vector | ***Brand and Perrimon, 1993*** | FLYBASE: FBmc0000383 | *Drosophila* Genomics Resource Center #1,000 |
| recombinant DNA reagent | pGL3-Rluc | This study | | See Materials and Methods; Dr. Nicholas Baker's lab |
| recombinant DNA reagent | p-GL3-H-T1 | This study | | See Materials and Methods; Dr. Nicholas Baker's lab |
| recombinant DNA reagent | p-GL3-H-T2 | This study | | See Materials and Methods; Dr. Nicholas Baker's lab |
| recombinant DNA reagent | p-GL3-H-T3 | This study | | See Materials and Methods; Dr. Nicholas Baker's lab |
| recombinant DNA reagent | p-GL3-H-T1S | This study | | See Materials and Methods; Dr. Nicholas Baker's lab |
| recombinant DNA reagent | p-GL3-H-T2S | This study | | See Materials and Methods; Dr. Nicholas Baker's lab |
| recombinant DNA reagent | pGL3-X-T1 | This study | | See Materials and Methods; Dr. Nicholas Baker's lab |
| recombinant DNA reagent | pGL3-X-T2 | This study | | See Materials and Methods; Dr. Nicholas Baker's lab |
| recombinant DNA reagent | pGL3-X-T3 | This study | | See Materials and Methods; Dr. Nicholas Baker's lab |

*Continued on next page*

*Continued*

| Reagent type (species) or resource | Designation | Source or reference | Identifiers | Additional information |
|---|---|---|---|---|
| recombinant DNA reagent | pGL3-X-T1S | This study | | See Materials and Methods; Dr. Nicholas Baker's lab |
| recombinant DNA reagent | pGL3-X-T2S | This study | | See Materials and Methods; Dr. Nicholas Baker's lab |

## Experimental animals

Fly strains were generally maintained at 25 °C on yeast cornmeal agar. Yeast-glucose medium was generally used for mosaic experiments (*Sullivan et al., 2000*). Sex of larvae dissected for most imaginal disc studies was not differentiated.

## Clonal analysis

Genetic mosaics were generated using the FLP/FRT system using inducible heat-shock FLP (hsFLP) transgenic strains. For making clones through mitotic recombination using inducible heat-shock FLP (hsFLP), larvae of $Rp^{+}$ genotypes were subjected to 10–20 min heat shock at 37 °C, 60 ± 12 hours after egg laying (AEL) and dissected 72 hr later. For making clones by excision of a FRT cassette, larvae were subjected to 10–30 min heat shock at 37 °C (details in *Supplementary file 2*), 36 ± 12 AEL for wild type background or 60 ± 12 hr AEL for $Rp^{+}$ background, and dissected 72 hr later.

## *Drosophila* stocks

Full genotypes for all the experiments are listed in *Supplementary file 2*. The following genetic strains were used: UAS-PPP1R15 (BL76250), UAS-PERK-RNAi (v110278 and v16427), UAS-Gcn2-RNAi (v103976), TRE-dsRED, P[GAL4-Act5C(FRT.CD2). P]S, P[UAS-His-RFP]3 (isolated from BL51308), UAS-TAF1B-RNAi (BL61957 and v105783), UAS-PPP1R15-RNAi (v107545 and BL 33011), UAS-w-RNAi (BL33623), UAS-CG6272-RNAi (BL33652), UAS-Xbp1-EGFP (BL60731), UAS-eIF4G-RNAi (v17002), UAS-eEF2-RNAi (v107268), UAS-eEF1α1-RNAi (v104502), UAS-eIF5Å-RNAi (v101513), UAS-eIF6-RNAi (v108094), UAS-Bsk[DN] (BL9311). Other stocks are described in *Lee et al., 2018*.

## Immunohistochemistry and antibody labeling

For most antibody labeling, imaginal discs were dissected from late 3rd instar larvae in 1xPBS buffer and fixed in 4% formaldehyde in 1 x PEM buffer (1xPEM:100 mM Pipes, 1 mM EGTA, 1 mM MgCl2, pH 6.9). For p-eIF2α and p-RpS6 detection, larvae were dissected in *Drosophila* S2 medium one by one and transferred immediately to fixative. Fixed imaginal discs were 3 x washed in PT (0.2% Triton X-100, 1xPBS) and blocked for 1 hr in PBT buffer (0.2% Triton X-100, 0.5% BSA, 1 x PBS). Discs were incubated in primary antibody in PBT overnight at 4 °C, washed three times with PT for 5–10 min each and incubated in secondary antibody in PBT for 3–4 hr at room temperature, and washed three times with PT for 5–10 min. After washes, samples were rinsed in 1 x PBS and the samples were incubated with the NuclearMask reagent (Thermofisher, H10325) for 10–15 min at room temperature. After washing 2 x with 1 x PBS the imaginal discs were mounted in VECTASHIELD antifade mounting medium (Vector Laboratories, H-1000). In experiments that we wanted to parallel process control samples on the same tube (e.g. *Figure 5C* vs 5 J), we used male parents that had the genotypes hsFLP; TRE-dsRed/(PPP1R15 or Xrp1RNAi or PERKRNAi); act>> Gal4, UAS-GFP and cross them with females from the RNAi of interest. The genotypes in the same tube were discriminated using dsRed before the addition of the secondary antibody. We used the following antibodies for staining: rabbit anti-phospho-RpS6 at 1:200 (1:200) (*Romero-Pozuelo et al., 2017*), rabbit anti-p62 (*Pircs et al., 2012*), rabbit anti-phospho-eIF2α at 1:100 (Thermofisher, 44–728 G, and Cell Signaling Technologies), rabbit anti-Xrp1 at 1:200 (*Francis et al., 2016*), mouse anti-b-Galactosidase (J1e7, DSHB), rabbit anti-GFP, rabbit anti-active-Dcp1 (Cell Signaling Techonology Cat#9578, 1:100), Y10b(1:100) (Thermofisher, MA1-13017), RpL9(1:100)(Abcam, ab50384),rabbit-anti-Rack1 (1:100) (Cell Signalling, D59D5), rabbit anti-hRpL10Ab (1:100) (Sigma, Cat# SAB1101199). Secondary Antibodies were Cy2- and Cy5- conjugates (Jackson Immunoresearch) or Alexa Fluor conjugates (Thermofisher). Previous experiments established that significant results could be obtained from five replicates, although many

more were imaged in most cases. No calculations regarding sample sizes were performed. No outliers or divergent results were excluded from analysis.

## Image acquisition and processing

Confocal laser scanning images were acquired with a Leica Laser scanning microscope SP8 using 20 x and 40 x objectives. Images were processed using Image J1.44j and Adobe Photoshop CS5 Extended. Thoracic bristle images were recorded using Leica M205 FA and Leica Application Suite X.

## Measurement of in vivo translation

Translation was detected by the Click-iT Plus OPP Alexa Fluor 594 or 488 Protein Synthesis Assay Kit (Thermofisher, C10457) as described earlier (*Lee et al., 2018*). Larvae were inverted in Schneider's *Drosophila* medium (containing 10% heat inactivated Fetal Bovine Serum, Gibco) and transferred in fresh medium containing 1:1000 (20 µM) of Click-iT OPP reagent. Samples were incubated at room temperature for 15 min and rinsed once with PBS. The samples were fixed in 4% formaldehyde in 1 x PEM buffer (100 mM Pipes, 1 mM EGTA, 1 mM MgCl2) for 20 min, washed once with 1 x PBS and subsequently washed with 0.5% Triton in 1 x PBS for 10 min and then incubated for 10 min with 3% BSA in 1 x PBS. The Click reaction took place in the dark at room temperature for 30 min. Samples were washed once with the rinse buffer of the Click reaction kit, 2 min with 3% BSA in 1 x PBS, incubated for 1 hr at room temperature with PBT (1 x PBS, 0.2% Triton, 0.5% BSA) and after that incubated overnight with the primary antibodies at 4°C. Samples were washed 3 x with PT buffer (1 x PBS, 0.2% Triton) and the secondary antibody was added for 2 hr in room temperature. After 3 x washes with PT and 1 x with 1 x PBS, the samples were incubated with the Nuclear Mask reagent (1:2000) of the Click-iT kit for 30 min. After washing 2 x with 1 x PBS the imaginal discs were mounted in Vectashield. Confocal laser scanning images were acquired with a Leica Laser scanning microscope SP8.

## Northern analysis

RNA extraction, northern blotting procedures, and 18 S, 5.8 S, tubulin and actin probeswere as described (*Lee et al., 2018*). Previous studies established that significant results could be obtained from three biological replicates. A biological replicate represents an independent RNA isolation, gel, and blot experiment.

The following primers were used to amplify the new probes in this paper:

ITS2 probe:

> 5'- CTTTAATTAATTTTATAGTGCTGCTTGG-3'
> 5'- TAATACGACTCACTATAGGGTTGTATATAACTTTATCTTG-3'
> 28 S probe:
> 5'-GCAGAGAGATATGGTAGATGGGC -3'
> 5'- TAATACGACTCACTATAGGGTTCCACAATTGGCTACGTAACT-3'

ITS1 probe:

> 5'- GGAAGGATCATTATTGTATAATATC-3'
> 5'- TAATACGACTCACTATAGGGATGATTACCACACATTCG-3'
> 7SL probe:
> 5'- TCGACTGGAAGGTTGGCAGCTTCTG-3'
> 5'- TAATACGACTCACTATAGGGATTGTGGTCCAACCATATCG-3'

## Plasmid cloning

All the new plasmids described below were confirmed by DNA sequencing.

Control *Renilla* luciferase plasmid: The pGL3-Promoter Vector (Promega) was modified by replacement of the SV40 promoter by the *Drosophila* actin promoter from the pAct5.1/V5-His C vector (Thermo Scientific), and the firefly luciferase coding sequence by the Renilla luciferase (RLuc) coding sequence from the pIS1 plasmid (Addgene), yielding the pGL3-Rluc plasmid.

*Firefly* luciferase plasmids: The SV40 core promoter of the pGL3-Promoter Vector was by hsp70 and Xrp1core promoters, amplified from the pUAST vector (*Drosophila* Genomics Resource Center) and from wild-type *Drosophila* genomic DNA respectively, using primers with XhoI and HindIII restriction sites. The resulting pGL3-H and pGL3-X plasmids were digested with Xho1 for insertion of annealed

complementary oligonucleotides containing multiple copies of Target 1, Target 2, Target 3, or shuffled Target one or Target two sequences, resulting in the p-GL3-H-T1, p-GL3-H-T2, p-GL3-H-T3, p-GL3-H-T1S, p-GL3-H-T2S, pGL3-X-T1, pGL3-X-T2, pGL3-X-T3, pGL3-X-T1S, and pGL3-X-T2S plasmids.

Inducible expression plasmids: The Xrp1 (with and without its 3'UTR sequence) and Irbp18 (CG6272) coding regions were amplified from pUAST-Xrp1-HA and pUAST-CG6272 (*Blanco et al., 2020*), and inserted into pMT/V5-His A (Thermo Scientific) usingXhoI and SpeI target sites, resulting in three inducible protein plasmids: pMT-Xrp1$^{HA}$Δ3'UTR, pMT-Xrp1$^{HA}$ and pMT-Irbp18$^{V5/His}$. pMT-Xrp1$^{HA}$ was not used further as it did not express Xrp1 protein in S2 cells.

## S2 cell culture and luciferase assays

*Drosophila* S2 cells from the *Drosophila* Genomics Resource Center (DGRC - stock#6) were cultured in Schneider's medium (Thermo Scientific) supplemented with 10% Heat-Inactivated Fetal Bovine Serum (Thermo Scientific) at 25 °C following the *General procedures for maintenance of Drosophila cell lines* from the DGRC. For luciferase assays, S2 cells were plated in 24-well plates, 5 × 10$^5$ cells per well. After 24 hr cells were transfected with the indicated combination of control Rluc (0.15 ng/well), protein expression (15 ng/well) and target (4.5 ng/well) plasmids using TransIT-2020 Transfection Reagent (Mirus) following the manufacturer's instructions. After 24 hr, copper sulfate was added to a final concentration of 0.35 mM. After a further 24 hr cells were lysed and *Renilla* and *Firefly* luciferases' activity measured with a luminometer, following the instructions from the Dual-Luciferase Reporter Assay System (Promega). *Firefly* signal was normalized to the internal *Renilla* control. Each transfection was performed in triplicate, and experiments performed independently at least three times.

## mRNA-Seq Analysis

In order to interrogate the RNA-Seq data (GSE112864 and GSE124924)(*Lee et al., 2018*; *Ji et al., 2019*) for the presence and abundance of transposons, we firstly retrieved a list of the known *Drosophila melanogaster* transposons from FlyBase (https://flybase.org/) as well as the related FASTA sequences (version r6.41) for which a dedicated Bowtie2 index was constructed. Subsequently, we realigned the RNA-Seq FASTQ files to the transposons using Bowtie2 with default parameters, while restricting the output of unaligned reads (--no-unal option) for faster later quantification. After the alignment, a raw transposon read counts table was constructed using samtools. Final quantification was obtained with RPKM transformation using the RNA-Seq sample library sizes and the lengths of each transposon.

## Acknowledgements

We thank Don Rio, Juhász Gábor and Aurelio Teleman for antibodies and Dirk Bohman, Katerina Papanikolopoulou, Hyung Don Ryoo, Efthimios Skoulakis and Eleni Tsakiri for other reagents. We thank Christos Delidakis, Nikolaos Konstantinides, Amit Kumar, Sudershana Nair, Venkateswara Reddy,Efthimios Skoulakis, and Deepika Vasudevan for comments on an earlier version of the manuscript. MK wants to specially thank Efthimios Skoulakis for hosting her research activities. We thank Andreas Stasinopoulos for discussions, Tao Wang for statistical advice, and Hyung Don Ryoo for sharing unpublished results. This work was supported by NIH grant GM120451 to NEB. *Drosophila* stocks were obtained from the Bloomington *Drosophila* Stock Center and Vienna Stock Resource Center (supported by NIH P40OD018537). Some confocal microscopy was performed in the Analytical Imaging Facility of the Albert Einstein College of Medicine (supported by the NCI P30CA013330) using the Leica SP8 microscope acquired through NIH SIG 1S10 OD023591, as well as a Leica TCS SP8X White Light Laser confocal system at Alexander Fleming Institute supported by the BIO-IMAGING-GR MIS 5002755. Some data in this paper are from a thesis submitted in partial fulfillment of the requirements for the Degree of Doctor of Philosophy in the Biomedical Sciences, Albert Einstein College of Medicine.

# Additional information

## Funding

| Funder | Grant reference number | Author |
|--------|------------------------|--------|
| National Institute of General Medical Sciences | research project grant GM120451 | Nicholas E Baker |
| NIH Office of the Director | instrumentation grant S10OD023591 | Nicholas E Baker |

The funders had no role in study design, data collection and interpretation, or the decision to submit the work for publication.

## Author contributions

Marianthi Kiparaki, Conceptualization, Data curation, Formal analysis, Investigation, Methodology, Validation, Visualization, Writing – original draft, Writing – review and editing; Chaitali Khan, Conceptualization, Formal analysis, Investigation, Visualization, Writing – review and editing; Virginia Folgado-Marco, Conceptualization, Formal analysis, Investigation, Visualization, Writing – original draft; Jacky Chuen, Investigation; Panagiotis Moulos, Conceptualization, Data curation, Formal analysis, Investigation, Methodology, Visualization; Nicholas E Baker, Conceptualization, Data curation, Formal analysis, Funding acquisition, Investigation, Writing – review and editing

## Author ORCIDs

Marianthi Kiparaki (ID) http://orcid.org/0000-0002-0157-9064
Chaitali Khan (ID) http://orcid.org/0000-0001-7344-8605
Jacky Chuen (ID) http://orcid.org/0000-0003-4781-6907
Panagiotis Moulos (ID) http://orcid.org/0000-0002-4199-0333
Nicholas E Baker (ID) http://orcid.org/0000-0002-4250-3488

## Decision letter and Author response

Decision letter https://doi.org/10.7554/eLife.71705.sa1
Author response https://doi.org/10.7554/eLife.71705.sa2

# Additional files

## Supplementary files

- Supplementary file 1. mRNA-seq data for transposable elements.
- Supplementary file 2. Genotypes, numbers, and heat-shock parameters for figures.
- Transparent reporting form

## Data availability

mRNA-Seq data were analyzed from datasets available from GEO with accession numbers GSE112864 and GSE124924. All other data generated or analysed during this study are included in the manuscript and supporting files. Source data files have been provided for Figure 1, Figure 2, Figure 2-figure supplement 1, Figure 8-figure supplement 4, Figure 10 and Figure 10-figure supplement 1.

The following previously published datasets were used:

| Author(s) | Year | Dataset title | Dataset URL | Database and Identifier |
|-----------|------|---------------|-------------|-------------------------|
| Kiparaki M, Blanco J, Folgado V, Ji Z, Kumar A, Rimesso G, Baker N | 2018 | RNA-seq analysis to assess transcriptional effects of Rp mutations in wing imaginal discs and their dependence on Xrp1 | https://www.ncbi.nlm.nih.gov/geo/query/acc.cgi?acc=GSE112864 | NCBI Gene Expression Omnibus, GSE112864 |

*Continued on next page*

*Continued*

| Author(s) | Year | Dataset title | Dataset URL | Database and Identifier |
|---|---|---|---|---|
| Baker N, Kiparaki M, Blanco J, Folgado V, Ji Z, Kumar A, Rimesso G | 2019 | mRNA Seq analysis of *Drosophila* wing imaginal discs from Rp mutants and controls in the presence and absence of RpS12 mutations RpS12 | https://www.ncbi.nlm.nih.gov/geo/query/acc.cgi?acc=GSE124924 | NCBI Gene Expression Omnibus, GSE124924 |

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
