## [Editor Report]

This paper will be of broad interest to biologists and has potential clinical relevance. It reveals that defects in ribosome biogenesis or function lead to PERK-phosphorylation of eIF2alpha and render cells vulnerable to cell competition by wild-type neighbors. Cell competition requires induction of the transcription factor Xrp1 in the mutant cell. Thus, the authors demonstrate that Xrp1 is the master regulator of cell competition. A series of compelling experimental manipulations dissecting the epistatic relationship between ribosome defects, Xrp1, eIF2alpha phosphorylation and cell competition support the key claims of the paper.

---

## [Decision Letter]

**Decision letter after peer review:**

Thank you for submitting your article "The transcription factor Xrp1 orchestrates both reduced translation and cell competition upon defective ribosome assembly or function" for consideration by *eLife*. Your article has been reviewed by 3 peer reviewers, including Erika A Bach as Reviewing Editor and Reviewer #1, and the evaluation has been overseen by James Manley as the Senior Editor.

Essential revisions:

Experimental:

1. A key conclusion of the paper is that "interrupting the translation cycle activates Xrp-1 dependent cell death independently of diminished translation". Most of the data supporting this conclusion are contained in Figure 7 and its supplement, and some of these images are not compelling. Specifically, in Figure 7A,D,G the GFP-positive UAS-PPP1R15, UAS-eEF2RNAi loser clones are very large, encompassing most of the field of the of view. In these clones, p-eIF2alpha is not upregulated. However, in Figure 7D,C,E,F,H,I, these same clones in the same genetic background are very small and are obviously being outcompeted. Why is there such a huge discrepancy in clone size across panels? Furthermore, the level of p-eIF2alpha was not monitored in other depletions of translation factors in Figure 7, Supplement 1, panels A-D. First, the authors need to provide better examples for p-eIF2alpha in Figure 7A,D,G. Second, they need to provide representative examples of p-eIF2alpha in Figure 7, Supplement 1, panels A-D. Third, they need to provide some quantification of the p-eIF2alpha results in Figure 7A,D,G and Figure 7, Supplement 1, panels A-D. Fourth, they need to provide representative examples of OPP in Figure 7, Supplement 1, panels A-D.

2. The authors need to quantify some of the results, including (a) p-eiF2alpha results mentioned in point #1; (b) penetrance of Rp/+ phenotypes; (c) northern blot results in Figure 2.

3. The authors need to prove that larger p-eiF2alpha spots in discs are dividing cells by co-staining with pHH3.

4. They need to supply missing controls. For Figure 3, please supply WT clones in a wild-type background treated with OPP and p-eiF2alpha antibody. For Figure 7, please supply UAS-PPP1R15 flip-out clones alone labelled with p-eiF2alpha, OPP and Dcp-1.

5. They need to prove with higher magnification that Xrp1 is excluded from the nucleolus.

6. They need to test Ire1 activation (i.e., Xbp1-GFP) and GstD1-GFP induction in Rp/+ clones, not just in Rp/+ heterozygous backgrounds.

7. The authors need to clarify in the manuscript how competition can be independent of translation because their data clearly show that when competition occurs, the loser clone has *both* elevated Xrp1 and lower translation. While the authors conclude that being a loser is determined solely by upregulation of Xrp1 and is independent of relative translation rate, in every case they show where there is competition, the translation rate is lower in the loser clone (see Figure 7B,E,H). In some of those manipulations, when they remove Xrp1, the same mutant can have a higher or unchanged translation rate as wild-type (Figure 3, Supplement 1 panel G', L', Figure 5K', Figure 7T'). The authors' work does demonstrate that having differences in translation alone (without Xrp1 differences) cannot induce cell competition. However, their work does *not* show that differences in Xrp1 without differences in translation can cause competition.

8. The authors need to discuss their new findings in the context of previously published work on how Xrp1 impacts translation. While the reviewers are aware that overexpression of Xrp1 is lethal (Baillon et al., Sci Reports 2018), the authors do not discuss the fact that Xrp1 translation is *induced* in loser cells where overall translation is *lower*. The reviewers agree that figuring out the mechanism by which Xrp1 is induced under restrictive translation conditions is outside the scope of this work. However, given point 7 above, the authors must acknowledge these unknowns explicitly in the discussion.

Editorial – The manuscript lacks clarity in several places. Please modify the revised manuscript as per the specific reviewer's comments.

1. The abstract and the introduction need revision (see R2).

2. The narrative describing the results in Figure 7 that are essential to the paper are not described in sufficient detail. Please expand the text in lines 426-445 to include the rationale for the experiments, the observations and the conclusions.

3. In the Discussion, the authors should discuss the observation that p-eiF2alpha is both upstream and downstream of Xrp1 and whether there is a feedback loop?

4. The authors should speculate in the Discussion about what is downstream of Xrp1 that causes WT neighbors to outcompete less competitive cells.

5. The authors need to address whether they have previously shown that clonal over-expression of Xrp1 causes cell competition without any other perturbation. If this experiment has been done in a previous publication from the lab, it needs to be discussed in the context of the model.

6. The authors need to discuss in a few sentences why increased Xrp1 expression occurs throughout the "loser" clone but competitive death only occurs at the clone boundaries.

7. The authors should soften the conclusion of reduction of LSU levels in Rp/+ because they tested only two subunits. The same request is made for the effect of LSUs on proteotoxicity (line 334).

8. The authors should explain the seemingly contradictory observations that Dcp1 is observed outside and inside clone boundaries (Figure 6F', J') and also within clones (Figure 7 supplement 2B', D', J') but the model holds that competitive death should be at the boundaries.

9. The authors should acknowledge that p-eIF2alpha is not a constant entity and can vary in non-stress conditions like circadian rhythm.

10. Many typos need to be corrected.

Figure labeling:

1. Genotypes need to be better labelled in some figures (like Figure 1). The authors should avoid using the terms *labelled* and *unlabeled* and instead refer to the cells in mosaic discs as white or black. Every figure panel should have written on it some version of the genotype.

2. Genotypes should be in the figure legends themselves, in addition to Table S2. In other words, it should be easy for the reader to know the genotype of the labelled (white) cells and the unlabeled (black) cells.

3. Many figures need arrows, etc, to help the reader under the data.

4. In all of the source data excel files, please use first row to indicate the relevant figure to which these data refer.

*Reviewer #1 (Recommendations for the authors):*

1. Phospho-eiF2alpha is both upstream and downstream of Xrp1 – can the authors please comment on this in the Discussion. Is this a feedback loop?

2. Can the authors speculate in the Discussion about what is downstream of Xrp1 that causes WT neighbors to outcompete less competitive cells? For example, do the authors think that there is a cell surface receptor recognized by WT cells? Something else.

3. The authors need to quantify phospho-eiF2alpha results. At a minimum, they should report in a supplementary table the number of clones scored, the number of wing discs scored, and whether the clone boundary always coincides with the phospho-eiF2alpha result.

4. The authors need to prove that the larger phosphor-eiF2alpha spots in discs are dividing cells. They need to co-stain with phospho-Histone H3.

5. Reporting on penetrance of Rp/+ phenotypes – the authors need to report the number of discs examined and whether all discs/clones of the same genotype displayed similar results.

6. Genotypes need to be better labelled in some figures (like Figure 1). and should be in the figure legends themselves, in addition to Table S2. It should be easy for the reader to know the genotype of the labelled (white) cells and the unlabeled (black) cells. In fact, the authors should avoid using the terms labelling and unlabeled and instead refer to the cells in mosaic discs as white or black.

7. In all of the source data excel files, please use first row to indicate the relevant figure to which these data refer.

8. Figure 3 is missing controls. Please add to this figure panels of WT clones in a wild-type background treated with OPP and phospho-eiF2alpha antibody.

9. There is a discordance between lines 295-297 and lines 1013-1014. The former states that phosphor-eiF2alpha levels were decreased in Rp+/- cells when Xrp1 was heterozygous, white the latter states the opposite. This needs to be clarified.

10. Perk mRNA is up 1.5x, but this may not mean 1.5x protein. Is there a reliable PERK antibody to use to test this?

11. Figure 7 – please supply panels in this figure of missing control PPP1R15 flip-out clone alone labelled with p-eiF2alpha, OPP and Dcp-1.

12. P values greater than 1: In several places there are Padj values that are greater than 1. This is not possible. Probability cannot be greater than 1. This problem occurs on Lines 863, 866, 1222, 1227. Please address this.

*Reviewer #2 (Recommendations for the authors):*

Overall, this study makes important progress in the field of cell competition as described in the public comments. I'd like to note that application of their results is limited to minute-mediated cell competition as of now, and also in lower organisms since there is no clear homolog of Xrp1 in higher species. I am cautiously enthusiastic about the publication of the study in *eLife* with some recommendations for improvement as listed below:

1. The manuscript suffers from lack of clarity, particularly in the abstract and introduction where it is unclear what known/published data are and what are the new findings. e.g. in the abstract: "The changes in ribosomal subunit levels observed are not sufficient for these effects, which all depend on the AT hook, bZip domain protein Xrp1." is a finding of this manuscripts but reads as though it is a previous known. There are also several grammatical errors that made the manuscript difficult to follow e.g. line 65 "the regulation of translation are important targets of cellular regulation" does not quite communicate effectively. I recommend that the authors consider revising linguistic aspects of the manuscript.

2. To strengthen the authors' conclusion that Xrp1 is required for cell competition due to reduced translation, it would be useful to test if this also holds true when there is a duplication of an Rp rather than loss of a copy of Rp.

3. The authors claim that "SSU components are generally reduced in RpS+/- cells and levels of SSU components are generally reduced in RpS+/- cells and RpL27A+/- cells, whereas LSU levels are only reduced in RpL27A+/- cells" (Line 217) is not fully supported by their data since they test only two LSU proteins (L27, L14) which differ from each other. This conclusion needs to be either toned down or more LSU minutes should be tested to derive a well-supported general principle. The same goes for the effect of LSUs on proteotoxicity (line 334).

4. According to the legend, the imaginal disc images in Figure 1 have been captured at different confocal planes- it is unclear why this was done and why maximum projections were not used to keep imaging consistent across samples.

5. To effectively conclude that levels of pre-rRNA increase when Xrp1 is mutated (line 247), the northern blots in Figure 2 need to be quantified. As presented, the differences are not appreciable particularly when normalized to tubulin.

6. The modulation of rRNA biogenesis intermediates using TAF1B knockdown to reduce levels of pre-rRNA is a sledge-hammer approach at best which has multiple secondary effects (admittedly including reducing translation as demonstrated by them)- thus does not shed much light on if the accumulation of the aberrant rRNA intermediates effects cell competition or is simply an artefact of the experiment.

7. As stated in the public review point 2, the staining with fibrillarin in Figure 2 is not of sufficient quality to conclude that Xrp1 has no nucleolar localization.

8. As stated in the public review points 3 and 5, the stress reporters Xbp1-GFP and GstD1-GFP need to be tested in Rp+/- clones in order to exclude or establish a role for Ire1 and Xrp1 in the induction of the respective reporters during cell competition. In the absence of these, the conclusions are speculative at best.

9. One of the novel findings of the manuscript is that translation differences caused by non-Rp factors can also lead to cell competition and they demonstrate this by depleting eIF2, eIF4G, eIF6, eEF2, eEF1α1, or eIF5A in clones. To the best of my knowledge, none of these factors show minute-like phenotypes or haploinsufficiencies. Further in some conditions of translation factor knockdowns, there is extensive Dcp1 staining well outside the clone boundaries (Figure 6F', J') and also within clones (Figure 7 supplement 2B', D', J'), which is contrary to competition-induced death at boundaries. It would be helpful if the authors explain these seemingly contradictory observations.

*Reviewer #3 (Recommendations for the authors):*

The model advocated by the authors predicts that a clone overexpressing Xrp1 should be able to cause cell competition without any other perturbation. If his experiment has been done in a previous publication from the lab, it needs to be discussed in the context of the model. It would also seem that increased Xrp1 expression occurs throughout the "loser" clone and we still need to understand why the death occurs at boundaries. The authors could add a few sentences about this in the discussion.

The main point I raised in the "public review" could be addressed by asking whether co-expression of Myc and Xrp1 in a FLP-out clone might convert cells from winners to losers. If this combination results in increased translation compared to wild-type cells but still results in their elimination , it would show clearly that reduced translation is not necessary for elimination by cell competition. (This experiment may not work).

[Editors’ note: further revisions were suggested prior to acceptance, as described below.]

Thank you for resubmitting your work entitled "The transcription factor Xrp1 orchestrates both reduced translation and cell competition upon defective ribosome assembly or function" for further consideration by *eLife*. Your revised article has been evaluated by James Manley (Senior Editor) and a Reviewing Editor.

The manuscript has been improved but there are some remaining issues that need to be addressed, as outlined below:

1. Rebuttal #1 refers to OPP labeling in Figure 8, Supplement 2 B,F,H,K but these images do not have OPP labeling. Please clarify.

2. Rebuttal #6. The motivation, experimental approach and results for Figure 5, Supplement 3 are not clear, but this is an important figure. An entire figure with many clone genotypes cannot be accurately described to a broad scientific audience in 2 sentences (lines 430-433). Please rewrite this section so that it clearly explains motivation, experimental approach, and what data specific results allowed you to conclude that eIP2alpha "hyper-phosphorylation" was not necessary for cell completion. Please also explain hyper-phosphorylation of eIF2alpha since it is the first time you are using this term.

3. Lines 496-497. Figure 8, figure supplement 3I – there is still cell death in this panel but it is inaccurately described as "Xrp1 depletion eliminated cell death". I suggest that you change this phrase to "strongly reduced" instead of "eliminated".

---

## [Author Response]

Essential revisions:Experimental:1. A key conclusion of the paper is that "interrupting the translation cycle activates Xrp-1 dependent cell death independently of diminished translation". Most of the data supporting this conclusion are contained in Figure 7 and its supplement, and some of these images are not compelling. Specifically, in Figure 7A,D,G the GFP-positive UAS-PPP1R15, UAS-eEF2RNAi loser clones are very large, encompassing most of the field of the of view. In these clones, p-eIF2alpha is not upregulated. However, in Figure 7D,C,E,F,H,I, these same clones in the same genetic background are very small and are obviously being outcompeted. Why is there such a huge discrepancy in clone size across panels? Furthermore, the level of p-eIF2alpha was not monitored in other depletions of translation factors in Figure 7, Supplement 1, panels A-D. First, the authors need to provide better examples for p-eIF2alpha in Figure 7A,D,G. Second, they need to provide representative examples of p-eIF2alpha in Figure 7, Supplement 1, panels A-D. Third, they need to provide some quantification of the p-eIF2alpha results in Figure 7A,D,G and Figure 7, Supplement 1, panels A-D. Fourth, they need to provide representative examples of OPP in Figure 7, Supplement 1, panels A-D.

First, we replaced Figure 8A,D,G with better examples (Figure 7 has become Figure 8). Secondly, we added images of p-eIF2a in Figure 8 Supplement 1 panels A, D, G, J, M, and Figure 8 Supplement 2 panels A, D, G, J. Thirdly, we quantified p-eIF2a levels from multiple genotypes from Figure 7 and 8. The results are shown in Figure 8 Supplement 3. Fourthly, we added representative examples of OPP labeling for the genotypes in Figure 8 supplement 1. These are shown in Figure 8 Supplement 1 panels B,E,H,K,N, and Figure 8 Supplement 2 panels B,E,H,K.

2. The authors need to quantify some of the results, including (a) p-eiF2alpha results mentioned in point #1; (b) penetrance of Rp/+ phenotypes; (c) northern blot results in Figure 2.

a) p-eIF2a levels have now been quantified as described in point #1 above; b) Rp/+ phenotypes were all 100% penetrant unless otherwise noted. Number of discs examined is now listed in Supplementary file 2. c) We removed the reference to changing rRNA levels, as we now think this might have another cause.

3. The authors need to prove that larger p-eiF2alpha spots in discs are dividing cells by co-staining with pHH3.

We found that a different antibody to p-eIF2a does not label mitotic cells, so we have not pursued this aspect of the labeling further. We mention this discrepancy between antibodies in the text, please see lines 1232-1236.

4. They need to supply missing controls. For Figure 3, please supply WT clones in a wild-type background treated with OPP and p-eiF2alpha antibody. For Figure 7, please supply UAS-PPP1R15 flip-out clones alone labelled with p-eiF2alpha, OPP and Dcp-1.

We added control wild type clones in a wild type background labelled for p-eIF2a (Figure 3 supplement 1M) and OPP (Figure 3 supplement 1N). We added PPP1R15 flip out clones (and control w RNAi clones) labelled for p-eIF2a, OPP, and cell death (Figure 8 figure supplement 1 panels A-F).

5. They need to prove with higher magnification that Xrp1 is excluded from the nucleolus.

It had not been our intention to imply that Xrp1 does not enter the nucleolus. Rather, another group has claimed that in wild type cells Xrp1 is found only in the nucleolus, then released into the nucleus in Rp mutants. Our intention was to show that Xrp1 is not concentrated in the nucleolus (or anywhere else in the cell, in wild type cells), not that it cannot enter the nucleolus. We rewrote the section about nucleolar Xrp1 to eliminate the confusion. Please see lines 300-306.

6. They need to test Ire1 activation (i.e., Xbp1-GFP) and GstD1-GFP induction in Rp/+ clones, not just in Rp/+ heterozygous backgrounds.

We added GstD-LacZ expression in Rp/+ wing discs containing clones depleted for Xrp1 as Figure 6 figure supplement 2. We were unable to obtain Rp+/- clones in the Xbp-1 background in time.

7. The authors need to clarify in the manuscript how competition can be independent of translation because their data clearly show that when competition occurs, the loser clone has both elevated Xrp1 and lower translation. While the authors conclude that being a loser is determined solely by upregulation of Xrp1 and is independent of relative translation rate, in every case they show where there is competition, the translation rate is lower in the loser clone (see Figure 7B,E,H). In some of those manipulations, when they remove Xrp1, the same mutant can have a higher or unchanged translation rate as wild-type (Figure 3, Supplement 1 panel G', L', Figure 5K', Figure 7T'). The authors' work does demonstrate that having differences in translation alone (without Xrp1 differences) cannot induce cell competition. However, their work does not show that differences in Xrp1 without differences in translation can cause competition.

We added an experiment that tests the requirement for global translation differences in cell competition as Figure 5 figure supplement 3. The data shows that translation differences are not required for cell competition. Please see lines 430-433.

8. The authors need to discuss their new findings in the context of previously published work on how Xrp1 impacts translation. While the reviewers are aware that overexpression of Xrp1 is lethal (Baillon et al., Sci Reports 2018), the authors do not discuss the fact that Xrp1 translation is induced in loser cells where overall translation is lower. The reviewers agree that figuring out the mechanism by which Xrp1 is induced under restrictive translation conditions is outside the scope of this work. However, given point 7 above, the authors must acknowledge these unknowns explicitly in the discussion.

We added discussion of how Xrp1 might be translated under conditions where most translation is shut down. Please see lines 718-720.

Editorial – The manuscript lacks clarity in several places. Please modify the revised manuscript as per the specific reviewer's comments.1. The abstract and the introduction need revision (see R2).

We revised the abstract and introduction significantly, including issues raised by the reviewer, and reviewed the rest of the manuscript also. We added commas to the sentence beginning “Ribosome biogenesis, and the regulation of translation, are important targets of cellular regulation…”. Please see lines 75-79.

2. The narrative describing the results in Figure 7 that are essential to the paper are not described in sufficient detail. Please expand the text in lines 426-445 to include the rationale for the experiments, the observations and the conclusions.

This request might refer to comment #9 by reviewer 2 that none of the translation factors examined is known to be haploinsufficient or show Minute-like phenotypes. We have now included the haplo-sufficient nature of the genes encoding these factors to the manuscript. Please see lines 441-448. This supports our initial decision to knock down these factors to inhibit translation independently of the Minute phenotype, rather than contradicting it.

3. In the Discussion, the authors should discuss the observation that p-eiF2alpha is both upstream and downstream of Xrp1 and whether there is a feedback loop?

We have now added an experiment that tests the contribution of positive feedback between Perk and Xrp1 to translation in Rp mutants. The data are shown in Figure 6, with related data in Figure 6 figure supplements 1 and 2. These data show that the potential feed-forward loop is not essential. Please see lines 413-429 of the article.

4. The authors should speculate in the Discussion about what is downstream of Xrp1 that causes WT neighbors to outcompete less competitive cells.

We propose the model that one or more transcriptional targets of Xrp1 are the cause of cell competition, and added new data that shows this includes repetitive elements (Figure 10P,Q, Figure 10 Supplement 3, Supplementary Table1). Please see lines 802810.

5. The authors need to address whether they have previously shown that clonal over-expression of Xrp1 causes cell competition without any other perturbation. If this experiment has been done in a previous publication from the lab, it needs to be discussed in the context of the model.

6. The authors need to discuss in a few sentences why increased Xrp1 expression occurs throughout the "loser" clone but competitive death only occurs at the clone boundaries.

We added another sentence about this to the discussion. Please see lines 808-810. There is not much to say, however, as this is one of the questions outstanding in the field.

7. The authors should soften the conclusion of reduction of LSU levels in Rp/+ because they tested only two subunits. The same request is made for the effect of LSUs on proteotoxicity (line 334).

We softened the conclusions regarding LSU levels by stating only that there might be differences between LSU and SSU mutations (lines 213-218, 235-241, 535-539), and similarly regarding LSU mutants and autophagy (lines 358-363).

8. The authors should explain the seemingly contradictory observations that Dcp1 is observed outside and inside clone boundaries (Figure 6F', J') and also within clones (Figure 7 supplement 2B', D', J') but the model holds that competitive death should be at the boundaries.

This appearance of non-autonomy was mostly due to variations in GFP level making the genotype of dying cells unclear in some of the panels. We adjusted these panels and replaced some that were confusing (Figure 7 and Figure 7 figure supplement 1). We added a new enlarged figure to make the genotypes of dying cells more clear (Figure 7 figure supplement 2). We have adjusted or replaced several panels that were misleading. The large majority of the cell death affects the mutant cells, there is not much death outside clone boundaries.

9. The authors should acknowledge that p-eIF2alpha is not a constant entity and can vary in non-stress conditions like circadian rhythm.

We now mention that proteotoxic stress is not the only factor affecting eIF2a phosphorylation level (line 315-318). It does not seem worth mentioning circadian rhythm specifically, since the changes we describe are spatial, not temporal.

10. Many typos need to be corrected.

We have proofread the revised manuscript, in particular rewriting the discussion significantly.

Figure labeling:1. Genotypes need to be better labelled in some figures (like Figure 1). The authors should avoid using the terms labelled and unlabeled and instead refer to the cells in mosaic discs as white or black. Every figure panel should have written on it some version of the genotype.

Genotype labels were changed in Figure 1 and its supplements. Every figure panel now has the genotypes indicated, except that we do not always replicate this where an extra panel is added to highlight one channel that has already been shown and labeled in a multichannel image. We replaced or qualified the terms ‘labeled’ and ‘unlabeled’ in all the figure legends.

2. Genotypes should be in the figure legends themselves, in addition to Table S2. In other words, it should be easy for the reader to know the genotype of the labelled (white) cells and the unlabeled (black) cells.

We made these changes

3. Many figures need arrows, etc, to help the reader under the data.

We added some arrows, eg in Figure 5 supplement 2.

4. In all of the source data excel files, please use first row to indicate the relevant figure to which these data refer.

We added these titles to the source data files

Reviewer #1 (Recommendations for the authors):[…]12. P values greater than 1: In several places there are Padj values that are greater than 1. This is not possible. Probability cannot be greater than 1. This problem occurs on Lines 863, 866, 1222, 1227. Please address this.

Adjusted P value apparently >1 arise because of the multiple testing correction. As the reviewer notes, P>1 is impossible, and the convention is to set these adjusted values = 1, as we have now done. Previously, we had just reported the exact output of the software package. Please see lines 1063, 1066, 1738, 1742, 1821.

[Editors’ note: further revisions were suggested prior to acceptance, as described below.]

1. Rebuttal #1 refers to OPP labeling in Figure 8, Supplement 2 B,F,H,K but these images do not have OPP labeling. Please clarify.

Our apologies, we meant to refer to Figure 8, Supplement 3 B,F,H,K.

2. Rebuttal #6. The motivation, experimental approach and results for Figure 5, Supplement 3 are not clear, but this is an important figure. An entire figure with many clone genotypes cannot be accurately described to a broad scientific audience in 2 sentences (lines 430-433). Please rewrite this section so that it clearly explains motivation, experimental approach, and what data specific results allowed you to conclude that eIP2alpha "hyper-phosphorylation" was not necessary for cell completion. Please also explain hyper-phosphorylation of eIF2alpha since it is the first time you are using this term.

We have expanded this section to describe Figure 5, supplement 3 more fully, and removed the phrase “hyper-phosphorylation”, as we had not intended to imply anything unique about the phosphorylation in this genotype. Please see lines 431-441.

3. Lines 496-497. Figure 8, figure supplement 3I – there is still cell death in this panel but it is inaccurately described as "Xrp1 depletion eliminated cell death". I suggest that you change this phrase to "strongly reduced" instead of "eliminated".

We changed this to “Xrp1 depletion eliminated or strongly reduced cell death”. Please see lines 504-506.